


# Spring water anomalies before two consecutive earthquakes (Mw 7.7 and Mw 7.6) in Kahramanmaraş (Türkiye) on 6 February 2023

Sedat İnan[1,*], Hasan Çetin[2], Nurettin Yakupoğlu[1]

[1]Department of Geological Engineering, Istanbul Technical University, Ayazağa, Istanbul, Türkiye, 34467

[2] Department of Geology, Çukurova University, Adana, Türkiye, 01330

*Corresponding Author: sedatinan@itu.edu.tr

**ABSTRACT**

Understanding earthquake phenomena is always challenging. Search for reliable precursors of earthquakes are important but requires systematic and long-time monitoring employing multi-disciplinary techniques. In search of possible precursors, we obtained commercially bottled spring waters dated before and after the earthquakes of 6 February 2023. Hydrogeochemical precursors have been detected in commercially bottled natural spring waters (Ayran Spring and Bahçepınar Spring) which are at a distance of about 100 km and 175 km from the epicenters of the Mw 7.7 and Mw 7.6 Kahramanmaraş (Türkiye) Earthquakes of 6 February 2023, respectively. The available water samples cover the period from March 2022 to March 2023. The pre-earthquake anomaly is characterized by an increase in electrical conductivity and major ions ($Ca^{2+}$, $Mg^{2+}$, $K^+$, Na+, $Cl^-$, and $SO4^{2-}$) compared to the background for Ayran Spring water samples. The pre-earthquake anomaly lasted for at least six months. The anomaly in major ions sharply declined and the ion content approached the background values about two weeks after the earthquakes. Although only 6.5 kilometers away from the Ayran Spring, the bottled water samples of the Bahçepınar Spring did not show any anomalies in electrical conductivity; therefore, the samples were not analyzed for ion content. Bahçepınar water is collected from shallow boreholes dug into alluvial deposits which we believe are decoupled from the basement rocks and this may be the reason for the lack of abnormal water chemistry prior to the earthquakes. This attests to the fact that sampling locations are very important in the detection of possible earthquake precursors. Results on the Ayran spring water samples indicate that spring water chemical anomalies of discrete samples may provide



valuable information on pre-earthquake crustal deformation. Monitoring of spring waters,
along with other monitoring techniques in a multidisciplinary network, and for a sufficiently
long time, could potentially enable obtaining reliable proxy indicators of pre-earthquake
crustal deformation.

**Keywords:** geochemical anomalies, spring water, earthquake precursors,
Kahramanmaraş earthquakes, East Anatolian Fault Zone (EAFZ), Türkiye

## 1. Introduction

Two devastating earthquakes (Mw 7.7 and Mw 7.6) struck the Kahramanmaraş area in
Southern Turkey on 6 February 2023; the earthquakes occurred about 9 hours apart. The
earthquakes caused devastation claiming more than 50,000 deaths; leaving behind
thousands injured and/or homeless. Earthquakes of destructive magnitudes (e.g., M>7)
are naturally expected to occur at plate boundary settings (Figure 1) and Kahramanmaras
province is at the junction of the East Anatolian Fault System (EAFS) and the Dead Sea
Fault System (DAFS). However, the reason why such natural events turn into disasters
is mainly due to a lack of preparedness. Where buildings are not built to be sufficiently
earthquake-resistant, monitoring of crustal deformation and searching for reliable pre-
earthquake signals become more important. This is obviously a big challenge for earth
scientists to overcome. Although there is still a long way to go on this front, the scientific
literature is full of scattered but promising and encouraging cases.
Earthquakes are complex natural phenomena and their predictions have been long
viewed as difficult, if not impossible (e.g., Geller et al., 1997). Geochemical observations
to identify earthquake precursors were initiated in the late 1960s (Rikitake, 1979; Wakita
1996). Reviewing twenty years of relevant data Turcotte (1991) concluded that large
earthquakes are not preceded by reliable seismic precursors. Moreover, Geller et al.
(1997) claimed that earthquakes can never be predicted. However, for the last few
decades, there have been numerous reports of ground-based anomalies preceding major
earthquakes. (including but not limited to Rikitake, 1979; Dobrovolsky et al., 1979;
Birchard and Libby, 1978; Hauksson, 1981; Wakita et al., 1988; Sultankhodhaev, 1984;


Thomas et al., 1986; Rikitake, 1987; Etiope et al., 1997; Bella et al., 1998; Virk and Singh,
1993; King et al., 1995; Planinic et al, 2004; Claesson et al., 2004; Hartmann and Levy,
2006; Papadopoulos et al., 2006; Uyeda et al., 2008; İnan et al., 2008; İnan et al., 2010;
İnan et al, 2012a,b,c; Skelton et al., 2014 and 2019; Barberio et al., 2017; Ouzounov et
al., 2021; Gori and Barberio, 2022; Xiang and Peng, 2023). Compiling a review of claimed
precursors, Cicerone et al. (2009) conducted a survey of published scientific literature on
earthquake precursors and concluded that precursory anomalies seem to be recorded
where there is modern instrumentation. İnan et al. (2010 and 2012a) provided hints to
select monitoring sites. Recently, Conti et al. (2021) have provided a short review of
ground-based observations before earthquakes
Hydro-geochemical anomalies observed nearby seismic events are generally interpreted
to be related to the alteration of the groundwater circulating system by the changes in the
crustal stress/strain before earthquakes and mixing of different aquifers (e.g., Scholz et
al., 1973; Nur, 1974; Sibson et al., 1975; Sugisaki et al., 1996; Tsunogai and Wakita,
1995; Toutain et al., 1997; Claesson et al., 2004; Pérez et al., 2008; ·İnan et al., 2010;
Grant et al., 2011; İnan et al., 2012c; Doglioni et al., 2014; Ingebritsen and Manga, 2014;
Skelton et al., 2014 and 2019; Barberio et al., 2017; Gori and Barberio, 2022; Xiang and
Peng, 2023). However, another different approach based on "stress-activated positive
hole currents" has been suggested to play a role in the development of physicochemical
pre-earthquake stress indicators (Freund, 2011; Paudel et al., 2018)
As suggested by Nur (1974) and later by Rikitake (1987) precursory phenomena may
have a common physical basis which Scholz et al. (1973) called the "Dilatation and water
diffusion (DWD) model". Roeloffs (1996) noted that with respect to earthquake hydrology,
mechanical and fluid-dynamic effects can be modeled using poroelasticity. More recently,
the DWD model has been explained further (e.g., Doglioni et al., 2014; Wang and Manga,
2021). However, other authors have proposed a fundamentally different approach
(Freund et al., 2006; Freund, 2008; Freund, 2011; Paudel et al., 2018) to study and
evaluate physicochemical pre-earthquake stress indicators. Until the mechanism
controlling pre-earthquake processes is fully understood, it is worth noting that the
success of any pre-earthquake stress indicators may be compromised by the ever-
present crustal heterogeneity, anisotropy, and/or crustal blocks (Areshidze et al., 1992;


Tansi et al., 2005; Sol et al., 2007; İnan et al., 2012a; Yu et al., 2023). Microplate and/or
block boundaries are obstacles to pre-earthquake strain to transfer from one block to the
other (İnan et al., 2012a; Yu et al., 2023).

A multi-disciplinary earthquake observation network (GPS, seismology, soil radon, and
spring water monitoring stations) was established in Kahramanmaras and surrounding
provinces along the fault zones (Adana. Hatay, Malatya, Elazığ, Bingöl) in 2007 under the
scope of the TURDEP Project (İnan et al. 2007). In the Kahramanmaras area, due to its
quiescence, also borehole tilt monitoring stations were established. Continuous
monitoring was continued until the middle of 2012 and valuable multi-disciplinary data
were collected. However, throughout these five years, no earthquake of significant
magnitude (e.g. M>6) occurred to test the usefulness of the monitoring network, the
project was terminated by the funding organization due mainly to a lack of vision. As a
result, the earth science community was caught unprepared when two consecutive
devastating earthquakes struck the area on 6 February 2023. No ground (except GPS
and seismology) monitoring station data were available to detect possible pre-earthquake
anomalies. However, following the Mw 7.7 and Mw 7.6 Kahramanmaraş earthquakes, we
searched for bottled spring waters to analyze in search of possible pre-earthquake
anomalies. This proved difficult as the water supply to the large community affected by
the earthquakes was quite limited and businesses providing bottled spring waters were
also mostly shut down. Finally, we were able to obtain commercially bottled water
samples (dated before and after the earthquake) from the Ayran and Bahçepınar springs
which are located within about a 6.5-kilometer distance in the Osmaniye Province. The
spring waters are about 100 kilometers and 175 kilometers from the epicenter of the first
(Mw 7.7) and the second (Mw 7.6) earthquakes, respectively (Figure 1B). In this study,
we conducted electrical conductivity (Ec) measurements on bottled waters, and based on
the Ec results, we selected samples for analysis of major ions in water in search of pre-
earthquake anomalies. The spring water samples cover the range from March 2022 to
March 2023.

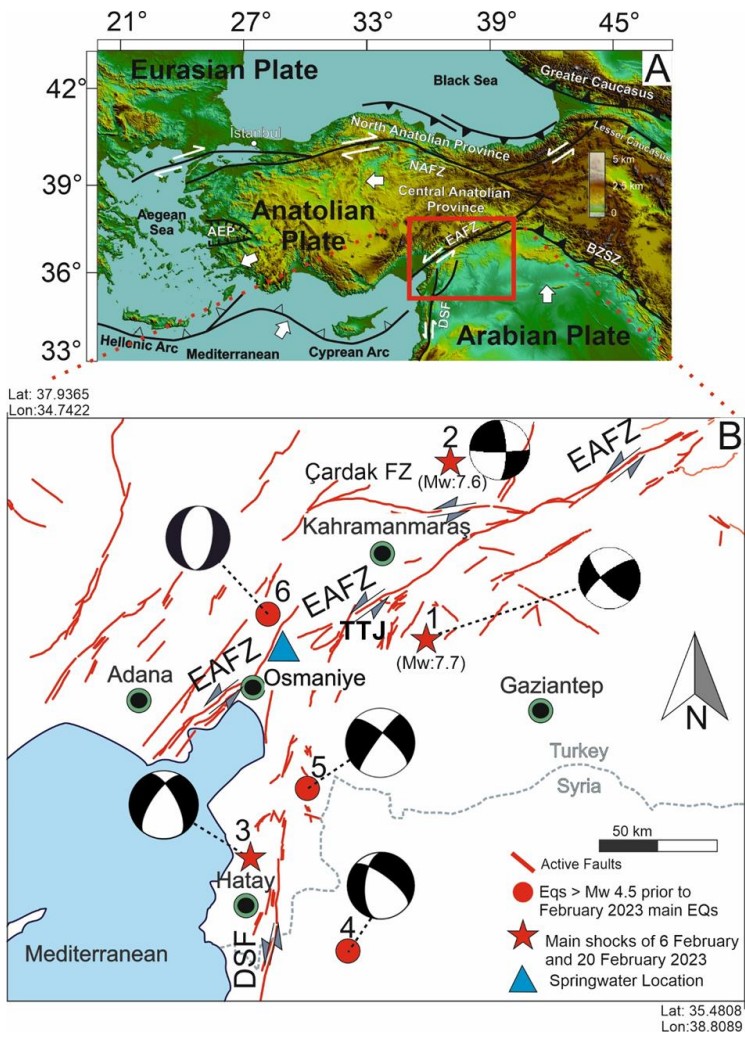


**Figure 1. A)** Neotectonics map of the Türkiye and surroundings (compiled from Sengör
and Yılmaz, 1981; Sengör et al., 1985; Hancock and Barka, 1987; Şaroğlu et al., 1992;
Barka and Reilinger, 1997; Bozkurt, 2001). **B)** Active fault map of the region affected by
the February 2023 Earthquakes (Perinçek and Çemen, 1990; Şaroğlu et al., 1992a; Cetin
et al., 2003). Red starts show the epicenters of the Mw 7.7 and 7.6 Kahramanmaraş
Earthquakes of 6 February 2023, and Mw 6.4 Hatay Earthquake of 20 February 2023.
Filled red circles show the locations of the earthquakes (Mw>4.5) that occurred in the
area (circle area with a radius of 150 km from the location of the water spring) between
September 2022 and 5 February 2023. Details of the earthquakes are given in Table 1.
TTJ is Türkoğlu Triple Junction. Beach balls are fault plane solutions of earthquakes and
were obtained from the Bogazici University Kandilli Observatory and Earthquake
Research Institute (KOERI) of Turkey; www.koeri.edu.tr



## 2. Active tectonics of the Kahramanmaraş region

Kahramanmaraş region takes place in the suture zone formed by the collision between Arabian and Anatolian plates (Figure 1A). After this collision, very important strike-slip fault zones were developed in the Anatolian plate due to the continuous northward movement of the Arabian plate and the resulting westward movement or escape of the Anatolian plate along two major fault zones, the North Anatolian Fault Zone (NAFZ) and the East Anatolian Fault Zone (EAFZ) (Ketin, 1948; McKenzie, 1972; Dewey and Şengör, 1979; Şengör and Yılmaz, 1981; Hempton, 1982; Şengör et al., 1985).

The East Anatolian Fault Zone (EAFZ) is approximately 550 km long, northeast-southwest trending, sinistral strike-slip fault (Figure 1A). It was first described by Allen (1969) and mapped by Arpat and Şaroğlu (1972). The EAFZ starts from Karlıova Triple Junction in the northeast, and it runs in the southwest direction, passes near the east-southeast of Kahramanmaraş, and joins another triple junction at Türkoğlu (TTJ in Figure 1B). The EAFZ then continues to the Hatay in the south direction to merge into the Dead Sea Fault Zone (DSFZ) (Allen, 1969; Arpat and Şaroğlu, 1972; Dewey and Sengör, 1979; Rotstein, 1984; Şengör et al., 1985; Kelling et al., 1987; Şaroğlu et al., 1992a and 1992b; Cetin et al., 2003; Yönlü et al., 2017). There are different interpretations, however, for the remainder of the fault zone after Türkoğlu Triple Junction (marked as TTJ in Figure 1B). Some studies extend the fault zone southwesterly to the Mediterranean Sea (McKenzie, 1972; Dewey et al., 1973; Jackson and McKenzie, 1984; Barka and Kadinsky-Cade, 1988; Karig and Kozlu, 1990; Kempler and Garfunkel, 1991; Westaway and Arger, 1996), joining it with the Cyprian Arc along which the convergence is taking place between the African and Anatolian plates (McKenzie, 1976; Dewey and Şengör, 1979). Others think that the fault zone ends around the TTJ (Lovelock, 1984; Chorowicz et al., 1994). According to Muehlberger and Gordon (1987), the EAFZ becomes the northern branch of the DSFZ

The seismicity of the study area is controlled by a complex interaction of the African, Arabian, and Eurasian plates (McKenzie, 1972). The seismicity of the EAFZ has been minimal for most of the last 100 years (Ambraseys, 1989). Historical earthquake records



show that Kahramanmaraş and its surroundings were affected by the two major
earthquakes in AD 1114 and AD 1513 (Soysal et al., 1981; Ambraseys, 1989). There had
been a long quiescence of more than 500 years in the Kahramanmaraş area before the
Mw 7.7 and Mw 7.6 earthquakes struck on 6 February 2023. About one year before these
earthquakes occurred, the area had been seismically quiet as suggested by only a few
M>4.5 earthquakes occurring in a circular area with a radius of 150 km; taking the Ayran
spring water as the center (Figure 1B and Table 1). The fault plane solutions (FPS) for
earthquakes #3, #4, and #5 suggest mainly normal faulting, whereas, for others
(earthquakes #1, #2, and #6), FPS suggest movement on dominantly left lateral strike-
slip faults (Figure 1B) as expected for left-lateral strike-slip nature of the EAFZ.
**Table 1.** Earthquakes' time, magnitude, and locations as received from www.koeri.edu.tr.
Earthquakes #1, #2, and #3 are the earthquakes of February 2023. Earthquakes #4, #5,
and #6 are those that have occurred in the circular area (with a radius of 150 km from the
Ayran spring water location) between September 1st, 2022 and 5 February 2023. The
locations of these earthquakes are given on the map (Figure 1B).

| Earthquake # | Magnitude (Mw) | Date | Time (GMT) | Latitude | longitude |
|---|---|---|---|---|---|
| 1 | 7.7 | 06.02.2023 | 01:17 | 37.1757 | 37.0850 |
| 2 | 7.6 | 06.02.2023 | 10:24 | 38.0818 | 37.1773 |
| 3 | 6.4 | 20.02.2023 | 17:04 | 36.0713 | 36.1012 |
| 4 | 4.6 | 12.01.2023 | 20:40 | 35.5712 | 36.6723 |
| 5 | 4.9 | 18.12.2022 | 18:13 | 36.3978 | 36.4455 |
| 6 | 5.0 | 11.10.2022 | 15:48 | 37.3025 | 36.2403 |


## 3. Samples and methods

### 3.1. Spring water samples
The spring water samples were received in commercial polyethylene bottles and brought
to Istanbul Technical University Laboratory for electrical conductivity measurements and
major ion analyses. Some of the samples had been bottled up to several months before
the analyses. However, this does not create any concern because much longer storage
in this kind of bottle has been reported to be appropriate in terms of keeping reliable
concentrations (Tsunogai and Wakita, 1995; İnan et al., 2012c).


The spring water samples cover the range from March 2022 to March 2023. It is worth
noting that the oldest sample predating the earthquakes was AYR 1 (dated 8 March 2022)
from the Ayran Spring. Other bottled water samples we could obtain from both springs
were dated between September 2022 and March 2023. In fact, we could not obtain any
samples dated between 8 March and 14 September 2022. The samples from September,
October, and November 2022 are limited but from December 2022 to January 2023,
available samples are several per month (Table 2).
### 3.2.    Spring water analysis
We first screened the bottled water samples by conducting electrical conductivity (Ec)
measurements, and based on the results, we selected samples for analysis of major ions.
Samples of the AYR spring water were analyzed by ion chromatography as discussed by
Zeyrek et al. (2010). Briefly, the samples were filtered at 0.45 µm and split into two
portions before analysis using an ion chromatography instrument (Dionex ICS 1000).
Sodium carbonate and methane-sulfonic acid were used as eluents for anion and cation
analyses, respectively. For calibration, DIONEX Certified Reference Standards were
used. Deionized water with a resistance better than 18.2 Megaohm was used for the
preparation of all eluents. Repeated measurements ascertained that the analytical
uncertainties for all anions and cations were below 5%. Electrical conductivity (Ec)
measurements for the bottled Bahçepınar (BPN) spring water samples and both Ec and
Ion analysis results for the bottled Ayran spring waters are listed in Table 2.
### 3.3. Statistical analysis of the data
For the statistical treatment of the data on major ion contents of the water samples, we
calculated the weighted average (weighted compared to the analytical error for each
point) and computed the 2σ external error (2 × αe) from the following equation

$$\alpha_e = \frac{\sum_{i=1}^{n}\left(x_i - \bar{x}\right)^2 / \sigma_i^2}{(n-1)\sum_{i=1}^{n} 1 / \sigma_i^2}$$




where x is the average and σ the analytical error on each measured point. The 2σ external
error (α$_e$) considers the general variability of all datasets and the analytical error on each
point; thus, we obtained the total error envelope for the samples that we consider
representing background (from 15 February to 31 March 2023; see Table 2 and Figure
3B).

**3.4. Relation between earthquake magnitude, distance, and precursory duration**

Slightly different relations between earthquake magnitude, duration of a precursory
anomaly, and the distance of the monitoring site to the earthquake epicenter have been
proposed. Dobrovolsky et al. (1979) proposed a theoretical relation (D= $10^{0.43*M}$) between
earthquake magnitude and maximum epicentral distance at which geochemical
anomalies may be observed.  This relation assumes a homogenous and isotropic crust.
Where M is the earthquake magnitude and D is the distance in kilometers to the
earthquake epicenter. Rikitake (1987) noted a slightly different relation (log T = a + b*M;
where a and b are constants, T is the duration of anomaly and M is the magnitude of an
earthquake). Moreover, Sultankhodhaev (1984) also reported a relation, between
earthquake magnitude, the distance of the monitoring site to the earthquake epicenter,
and duration of precursory anomaly (log (DT) = 0.63 * M – b; where D is the distance in
km, T is the duration of a precursory anomaly in days, and M is earthquake magnitude; b
is a constant taken as 0.15. All of these three relations provide a helpful initial idea about
what to expect of precursory anomalies in terms of duration and distance to the
earthquake epicenter. İnan et al. (2008 and 2010) verified Dobrovolsky et al.'s (1979)
relation for medium-size earthquakes (M<5.3). Accordingly, for an earthquake of
magnitude 4.5, the maximum distance for detection of possible geochemical anomalies
in the Ayran Spring water will be about 100 km. For contingency, we took a 150 km radius
and listed in Table 1 the earthquakes with M> 4.5 occurring between September 2022
and 5 February 2023 in order to compare with the water geochemical data we obtained
in this study.



**4.  Results and Discussion**

The bottled water samples from the Bahcepınar (BPN) did not show any meaningful (e.g.,
significant) variations in electrical conductivity (Ec) values; varying in a narrow range
between 220 and 230 microsiemens/cm (Table 2). Therefore, these samples were not
analyzed for major ions content because change (increase/decrease) of major ions
contents is expected to result in Ec variation (İnan et al., 2010; İnan et al., 2012c).
However, the bottled water samples from the Ayran (AYR) spring showed major variations
in the Ec values; varying in range between 50 and 200 microsiemens/cm. Therefore, the
AYR samples were analyzed for major ions. Possible reasons for not detecting any
anomaly in the Ec measurements of the BNP spring water samples have been
investigated. The investigation suggests that the reason may be the geological
environment of the springs. The AYR spring water emanates from Middle-Upper
Ordovician age metamorphic rocks (Kardere Formation) made up of quartzite,
metasandstone, metasiltstone, and metashale (Usta et al., 2015 and 2017), whereas the
BPN spring water is collected from shallow boreholes dug into valley-filling Quaternary
age alluvial deposits that are underlain by ophiolite (Figure 2).  The alluvial deposits reach
a thickness of about two hundred meters and the water reservoir within the alluvium
deposit is fed by precipitation and a nearby Bekdemir stream flowing towards the alluvial
deposit. It is interesting that the streams disappear to the south; suggesting that the
stream (creek) water is captured by the alluvial deposit. Since the BPN water is collected
from shallow boreholes (less than 100 meters) dug into alluvial deposits, we believe that
the alluvial deposits are decoupled from the basement rocks (which undergo pre-
earthquake stress) and this may be the reason for the lack of anomaly in water chemistry
prior to the earthquakes. This testifies to the importance of adequate geological
knowledge of the area before sampling discrete geochemical samples (water or soil gas)
and/or continuous monitoring in search of pre-earthquake signals (İnan et al., 2008; İnan
et al., 2010).

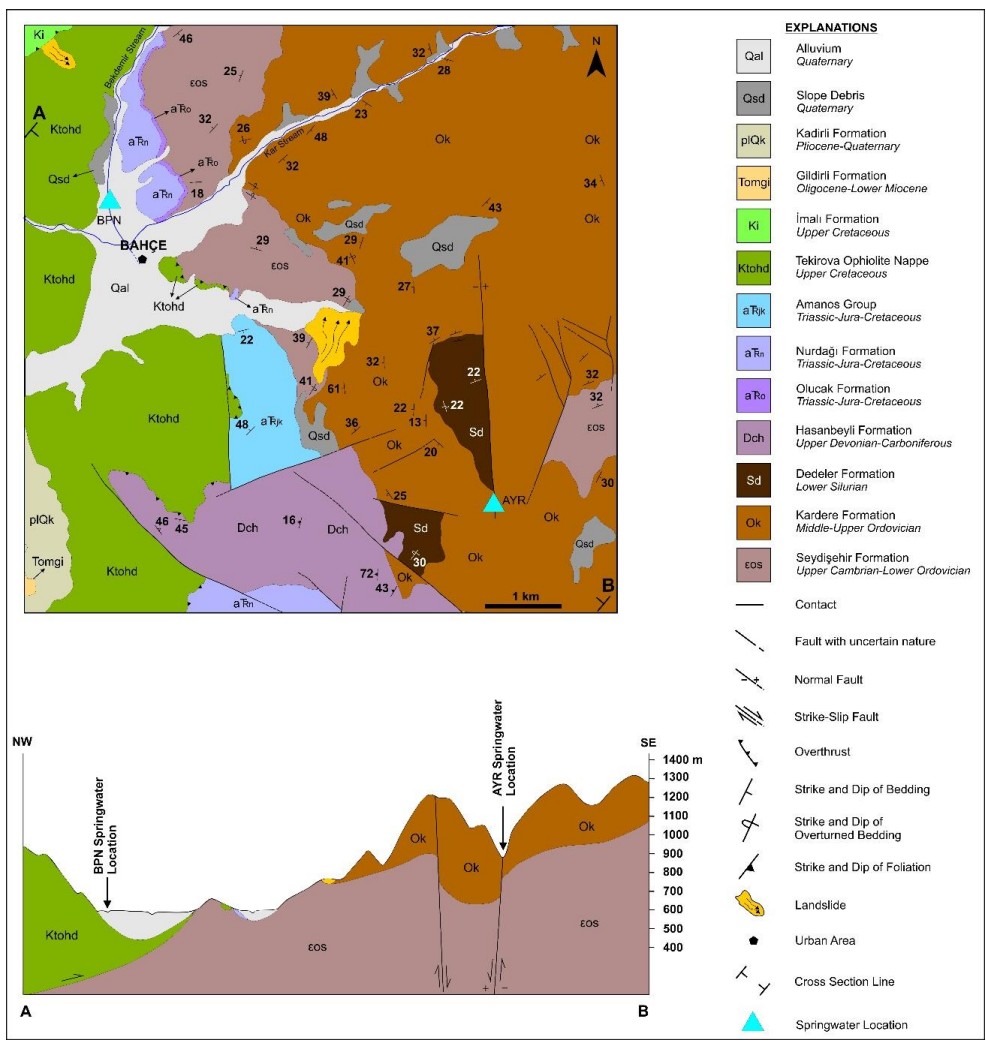


**Figure 2.** Locations and local geology of the water springs. (Modified from Usta et al., 2015 and 2017). The Ayran spring water emanates from a fault in the Metamorphic Kardere Formation (blue triangle shown at the lower right in the map) whereas the Bahçepınar spring water is obtained from the Quaternary Alluvium (blue triangle shown at the upper left in the map).


Variations of major ions in the AYR spring water samples are significant. It is clear that
pre-earthquake anomalies exceed the $\alpha_e$ (Figures 3B and Table 2). Before any
interpretation, we need to make sure that geochemical time series are not affected by
meteorological conditions. In this context, meteorological data have been obtained from





the Osmaniye Meteorology Station (located about 32 km SW of the AYR spring) and the
daily average air temperature and rainfall are shown in Figure 3c.
**Table 2.** Ec and major ion analysis results for the Ayran (AYR) and the Ec analysis results
for the Bahçepınar (BPN) bottled waters. The data for $Ca^{2+}$, $Mg^{2+}$, $K^+$, $Na+$, $Cl^-$, $SO4^{2-}$ for
the AYR samples are plotted in Figure 3B. Standard deviation (2σ) has been computed
considering cations/anions contents of samples dated from 15 February to 31 March
2023; the period which is considered to nearly represent background concentrations of
the water. These samples are marked in bold fonts.

| Sample ID | Date | $Cl^-$ | $SO_4^{-2}$ | $Na^+$ | $K^+$ | $Mg^{+2}$ | $Ca^{+2}$ | AYR Ec | Date | BPN Ec |
|---|---|---|---|---|---|---|---|---|---|---|
| AYR 1 | 08.03.2022 | 2.99 | 8.34 | 4.34 | 0.39 | 3.22 | 6.92 | 50 | 19.09.2022 | 220 |
| AYR 2 | 14.09.2022 | 7.37 | 13.08 | 12.10 | 1.08 | 7.73 | 17.54 | 150 | 07.11.2022 | 230 |
| AYR 3 | 06.10.2022 | 9.73 | 14.79 | 15.08 | 1.34 | 9.20 | 20.10 | 180 | 12.12.2022 | 230 |
| AYR 4 | 03.11.2022 | 9.99 | 15.52 | 15.66 | 1.39 | 9.50 | 20.72 | 170 | 19.12.2022 | 220 |
| AYR 5 | 13.12.2022 | 7.45 | 13.43 | 11.93 | 1.05 | 7.59 | 16.58 | 150 | 30.12.2022 | 220 |
| AYR 6 | 26.12.2022 | 11.06 | 17.35 | 16.49 | 1.56 | 10.19 | 22.19 | 190 | 08.01.2023 | 220 |
| AYR 7 | 29.12.2022 | 11.08 | 17.20 | 16.84 | 1.50 | 10.20 | 22.33 | 180 | 20.01.2023 | 220 |
| AYR 8 | 30.12.2022 | 10.97 | 17.29 | 16.78 | 1.50 | 10.17 | 22.28 | 190 | 24.01.2023 | 220 |
| AYR 9 | 03.01.2023 | 10.62 | 17.23 | 16.26 | 1.45 | 10.06 | 23.04 | 170 | 28.01.2023 | 220 |
| AYR 10 | 06.01.2023 | 11.12 | 17.56 | 16.91 | 1.49 | 10.29 | 22.51 | 190 | 04.02.2023 | 220 |
| AYR 11 | 11.01.2023 | 11.41 | 17.96 | 16.90 | 1.50 | 10.43 | 22.81 | 190 | 11.02.2023 | 220 |
| AYR 12 | 12.01.2023 | 11.60 | 18.21 | 17.22 | 1.53 | 10.50 | 22.99 | 200 | 17.02.2023 | 220 |
| AYR 13 | 27.01.2023 | 9.83 | 16.20 | 14.24 | 1.25 | 8.89 | 19.35 | 160 | 18.02.2023 | 230 |
| AYR 14 | 31.01.2023 | 11.04 | 17.62 | 15.81 | 1.39 | 9.87 | 21.58 | 180 | 02.03.2023 | 230 |
| AYR 15 | 01.02.2023 | 11.43 | 17.85 | 16.21 | 1.43 | 10.04 | 21.97 | 190 | 13.03.2023 | 220 |
| AYR 16 | 10.02.2023 | 9.09 | 15.59 | 13.29 | 1.16 | 8.46 | 18.33 | 180 | 22.03.2023 | 230 |
| AYR 17 | 12.02.2023 | 6.00 | 12.47 | 9.36 | 0.79 | 6.29 | 13.51 | 120 | | |
| AYR 18 | 13.02.2023 | 4.25 | 10.69 | 6.96 | 0.56 | 4.95 | 10.38 | 90 | | |
| **AYR 19** | **15.02.2023** | **3.54** | **10.30** | **6.28** | **0.50** | **4.79** | **9.65** | **80** | | |
| **AYR 20** | **16.02.2023** | **3.56** | **13.64** | **7.51** | **0.67** | **6.91** | **12.60** | **110** | | |
| **AYR 21** | **28.02.2023** | **3.29** | **10.54** | **5.79** | **0.47** | **4.60** | **9.23** | **80** | | |
| **AYR 22** | **02.03.2023** | **3.26** | **10.08** | **5.48** | **0.44** | **4.20** | **8.59** | **70** | | |
| **AYR 23** | **11.03.2023** | **3.36** | **9.85** | **5.49** | **0.43** | **4.21** | **8.62** | **70** | | |
| **AYR 24** | **13.03.2023** | **3.28** | **9.91** | **5.47** | **0.44** | **4.22** | **8.68** | **70** | | |
| **AYR 25** | **20.03.2023** | **3.28** | **9.96** | **5.36** | **0.43** | **4.22** | **8.73** | **70** | | |
| **AYR 26** | **24.03.2023** | **3.20** | **10.02** | **5.35** | **0.42** | **4.14** | **8.45** | **70** | | |
| **AYR 27** | **31.03.2023** | **3.31** | **10.13** | **5.40** | **0.43** | **4.16** | **8.47** | **70** | | |
| | *mean* | *0.37* | *1.13* | *0.60* | *0.05* | *0.46* | *0.94* | *7.78* | | |
| | Σ | *0.33* | *1.19* | *0.79* | *0.08* | *0.88* | *1.34* | *13.64* | | |
| | 2σ | *0.65* | *2.38* | *1.57* | *0.17* | *1.76* | *2.68* | *27.28* | | |
| | *mean +1* σ | *3.64* | *11.32* | *6.19* | *0.51* | *5.04* | *9.81* | *83.64* | | |
| | *mean +2* σ | *2.99* | *8.94* | *4.62* | *0.35* | *3.28* | *7.13* | *56.36* | | |


Air temperature gradually decreases from about 30°C in September 2022 to less than
10°C in February 2023 (Figure 3C). Daily rainfall is noticeably present in November 2022
and March 2023. Normally, variations in air temperature are not expected to affect the
chemical contents of the spring water (İnan et al., 2010 and 2012) but the effect of rainfall





on soil radon concentration is dominant (Inan et al., 2008, 2010, 2012b, Seyis et al.,
2022). All earthquakes listed in Table 1 are plotted on the meteorology time series in
Figure 3C and this shows that major and heavy rainfall took place right after the
devastating earthquakes of 6 February 2023. Based on the relatively low EC and low
major ion contents of the AYR spring water (Table 2) that is bottled and commercially
distributed, it can be said that this water is of shallow origin (Di Luccio et al., 2018). A
comparison of the geochemical time series and significant variations shown in Figure 3B
and the daily average rainfall data shown in Figure 3C reveals no correlations. İnan et al.
(2010 and 2012) compared meteorological time series with hydrogeochemical time series
and noted that meteorological conditions do not seem to play a role in water's major ion
contents. In this study, we compare rainfall data and geochemical time series (Figure 3)
and, as there is no correlation, we conclude that the increase of major ion contents
observed in AYR spring waters are not related to atmospheric variations (e.g., rainfall).
Therefore, it is safe to conclude that the chemical changes recorded in the spring water
must be related to crustal deformation associated with earthquake stress buildup.
As shown in Figure 3B, changes in the concentration of the major ionic species dissolved
in the AYR spring water were observed. Positive anomalies are recorded in the $Ca^{2+}$,
$Mg^{2+}$, $K^+$, $Na+$, $Cl^-$, and $SO4^{2-}$ contents (mg/l) before the 6 February Mw 7.7 and 7.6
Kahramanmaraş Earthquakes (Figure 3b; Table 2). These positive anomalies (increase
in dissolved ion content) started as early as September 2022; suggesting a pre-
earthquake anomaly of nearly six months. Considering Sultankhodhaev's (1984) relation
(log (DT) = 0.63 * M – b) between earthquake magnitude, precursory anomaly duration,
and the distance of the earthquake epicenter to the monitoring site, such a long duration
(six months) of a precursory anomaly we report in this study is very likely because the
magnitudes of the 7.7 and 7.6 devastating earthquakes are sufficiently big to cause such
a long precursory anomaly at a location about 100 km from the epicenter. Considering
the relation proposed by Sultankhodhaev (1984), such a magnitude of the earthquake
theoretically should lead to months-long of precursory anomaly in the geochemical
parameters at locations hundreds of kilometers far from the epicentral area.



In regard to changes in the dissolved ions in the AYR spring water, the following changes
are imminent.  The $Ca^{2+}$ and $Na+$ content increase (for the period between September
2022 to 15 February 2023) above the background by about 14 (mg/l) and 10 (mg/l),
respectively, and reach up to 22 (mg/l) and 16 (mg/l), respectively. This increase started
about six months before the 6 February earthquakes (EQ # 1 and EQ #2). Since we could
not obtain samples between 8 March 2022 and 14 September 2022, the anomaly could
have possibly started even earlier (any time between March and August 2022); so the
positive anomaly (e.g., increase) in the major ions started at least six months before the
6 February 2023 earthquakes. The $Mg^{2+}$ content also increased from about 4 (mg/l) to 10
(mg/) in the period September 2022 to 15 February 2023. Similar major increases were
also detected in $Cl^-$, and $SO4^{2-}$ contents. Water samples are relatively poor in $K^+$ content
therefore the increase, due to the scale of the graph, is not very obvious in Figure 3B.
However, the values given in Table 2 clearly indicate about four times an increase in the
$K^+$ content compared to the background concentrations (post-seismic samples collected
between February 15 and 31 March 2023).
The pre-earthquake anomaly in the AYR water samples is characterized by an increase
of up to 400% for the Ec and also major ions; namely $Ca^{2+}$, $Mg^{2+}$, $K^+$, $Na+$, $Cl^-$, and $SO4^{2-}$
before the 6 February 2023 Mw 7.7 and Mw 7.6 earthquakes (Figure 3B). Post-
earthquake samples show decreasing trends in all major ions. Analyses results of the
post-earthquake dated samples show that the spring water has had chemical stability
since the Middle February-Early March 2023; just two to three weeks after the
earthquakes (Figure 3B). We have also obtained a chemical analysis report on AYR water
submitted with the business license application of the company dated 29 August 2012.
The chemical analysis data of the samples collected more than 10 years ago include
values only for $Na+$, $Cl^-$, and $SO4^{2-}$ as 3.86, 3.12, and 8.37 mg/l, respectively. These
values are very close to the analysis result of the AYR water sample dated 8 March 2022
(AYR 1 which is the oldest sample in our data set) and the AYR water samples collected
after 15 February (Table 2); confirming that these samples represent background values
for the AYR spring water.



Immediately after the earthquake, the values started to decrease suggesting a reversible
chemical change (Figure 3B; Table 2). It is worth mentioning that the broad positive
anomaly detected in the AYR water chemistry (Figure 3B) that lasted for about six months
before the Mw 7.7 and Mw 7.6 earthquakes shows some transient decreases (about
Middle December 2022 and toward the end of January 2023). Following each transient
decrease, an increase in ion contents is observed and the broad positive anomaly
(starting from September 2022) is sustained until the date of the major earthquakes of 6
February 2023. The observations of sudden decrease and rebound in the major ion
contents of the water samples (taking place in Mid December 2022 and end of January
2023) may suggest sudden and short-lived crustal stress release related to smaller
earthquakes (e.g., EQ # 4 and EQ # 5). Soon after the major earthquakes (EQ # 1 and
EQ # 2), the major ion contents of the water samples show a sharp decline; almost
approaching the background values as early as 15 February 2023. One single positive
anomaly after the major earthquakes (EQ #1 and EQ #2) is detected in the sample dated
16 February 2023. The further increase of the ion contents of this sample seems to
suggest a short-term stress buildup prior to EQ # 3 (Mw 6.4) that occurred about 120 km
to the south of the Ayran Spring water location (Figure 1B). Considering Dobrovolsky et
al.'s (1979) theoretical relation ($R= 10^{0.43*M}$), an increase in major ions contents of the
Ayran Spring water is very likely to take place due to an earthquake of magnitude 6.4
occurring in 120 km distance.



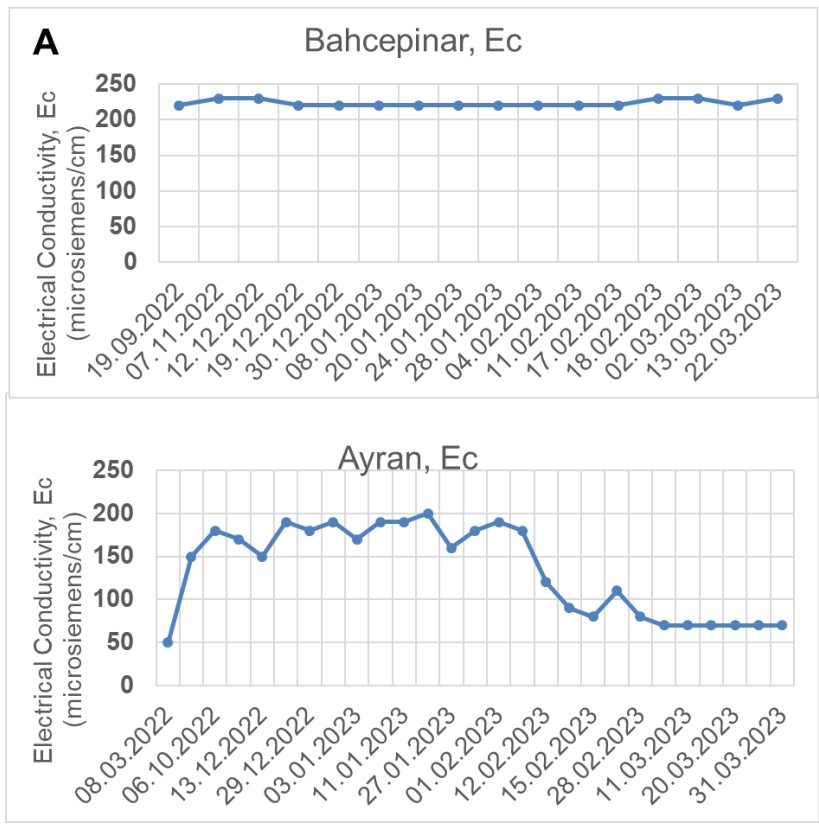


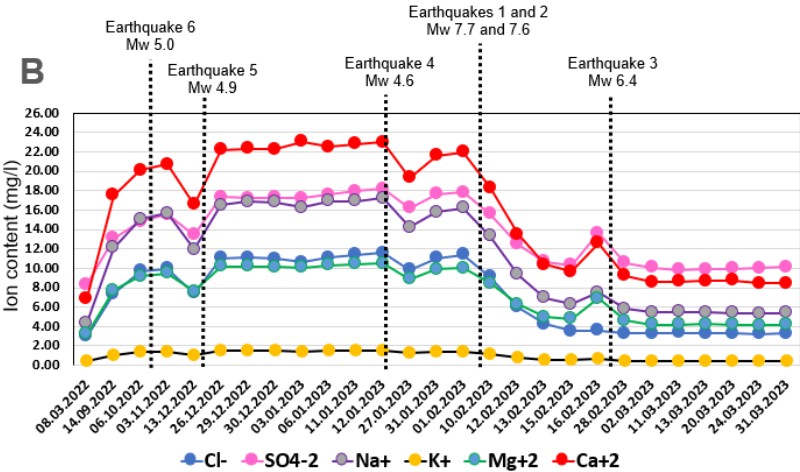

**Figure 3.** Time variation graphs of Ec for the Ayran (AYR) and the Bahçepınar (BPN)
bottled waters (**A**) and major ions for the AYR bottled waters (**B**). All data are listed in
Table 2.

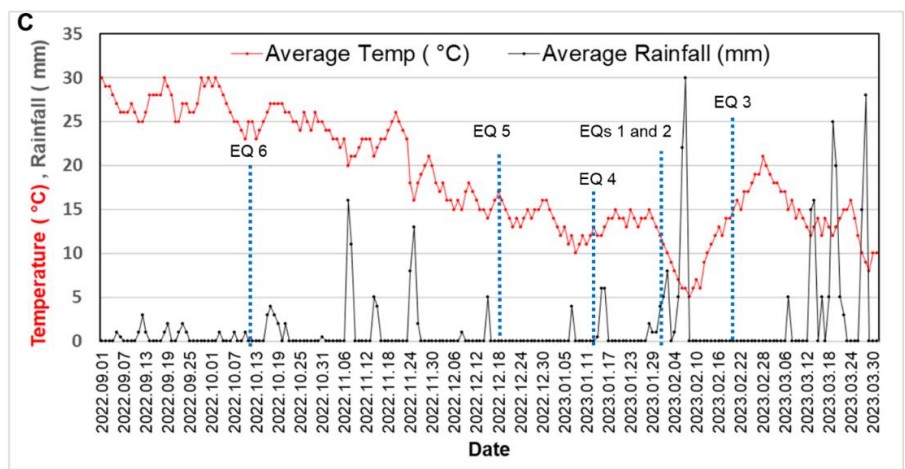

**Figure 3. Cont.** Daily average air temperature and rainfall at the Osmaniye meteorology station (37°07'N, 36°25'E; 32 km SW of the Ayran spring) between 1 September 2022 and 30 March 2023 (**C**).

(https://www.meteoblue.com/tr/hava/historyclimate/weatherarchive/
osmaniye_türkiye_303195). EQ1 through EQ6 are the earthquakes listed in Table 1.

We have shown and discussed the reliable precursory anomalies in the major ions of the bottled AYR spring water prior to the Mw 7.7 and Mw 7.6 earthquakes that occurred in the Kahramanmaraş region on 6 February 2023. However, the process(es) leading to the build-up of geochemical anomalies related to the earthquake cannot be inferred with certainty. However, some inferences based on previous observations can be made. For instance, Sibson (1992) suggested that extensive hydro-fracture dilatancy might develop prior to failure leading to the earthquake. Development of fractures probably enhances water circulation and mixing of different reservoirs leading to pre-earthquake anomalies (Italiano et al., 2004; Federico et al., 2008; İnan et al., 2010; İnan et al., 2012c; Skelton et al., 2014; Ingebritsen and Manga, 2014; Doglioni et al., 2014; Barberio et al., 2017; Skelton et al., 2019; Wang and Manga, 2021; Gori and Barberio, 2022;). Although the process(es) responsible for chemical anomalies detected in the Ayran spring waters prior to the 6 February 2023 earthquakes cannot be suggested with any certainty at this stage, two immediate mechanisms emerge: 1) a simple increase in fluid flow in the surrounding of the future epicenter and selective dissolution of some K–Mg–Ca-rich rocks (e.g.,


Federico et al., 2008); or 2) "electro-corrosion" whereby the dissolution of rocks is
accelerated by the flow of stress-activated positive hole currents (Balk et al., 2009;
Freund, 2011; Paudel et al., 2018). Following the second mechanism, the increased
content of major ions in water could be related to the oxidation of water to hydrogen
peroxide at the rock-water interface (Balk et al., 2009; Paudel et al., 2018). Freund (2011)
suggested that with the positive hole current flowing, the "corrosion" of the rock is
accelerated releasing into the water major cations and anions. Further work to be
conducted in this area may enable us to suggest the process(es) responsible for the pre-
earthquake geochemical anomalies we have discussed in the AYR spring water.

**5. Conclusions**

Hydrogeochemical precursors have been detected in commercially bottled water samples
of natural springs (Ayran Spring and Bahçepınar Spring) emanating from a location about
100 km distance from the epicenter of the Mw 7.7 Kahramanmaraş Earthquake of 6
February 2023. The pre-earthquake anomaly is characterized by an increase in $Ca^{2+}$,
$Mg^{2+}$, $K^+$, $Na+$, $Cl^-$, and $SO4^{2-}$ content in the bottled water samples of the Ayran spring.
Samples that are dated after the earthquakes (covering about two months after the
earthquake) show decreasing trends in all major ions. About three weeks after the
earthquake, the major ion contents of the spring water attained stability. At least six
months of pre-earthquake anomaly (increase) in the major ions content of the Ayran
spring water is imminent. It is worth noting that the Bahçepınar Spring water samples did
not show any anomalies in electrical conductivity therefore the samples were not
analyzed for ion content. Bahçepınar water is collected from shallow boreholes dug into
alluvial deposits which, we believe, are decoupled from the basement rocks and this may
be the reason for the lack of any significant change in the water chemistry prior to the
earthquakes. Here, we remind that geological knowledge of the investigated area and the
sampling site have paramount importance in sampling discrete samples for geochemical
analysis and/or conducting continuous monitoring. The results of this study suggest that
spring water chemical anomalies may be monitored as proxy indicators of pre-earthquake



crustal deformation. The physical mechanisms of the observed precursors are yet
impossible to explain with certainty at this stage. In order to be able to suggest the
mechanism(s) leading to the reported pre-earthquake geochemical anomalies, more work
needs to be conducted; especially multi-disciplinary (seismological, geodetical,
geochemical) and continuous earthquake monitoring networks must be established and
run for a sufficiently long time.

**Acknowledgements**
We appreciate all the technical help we have received on ion chromatography analyses
from Ms. Sevde Korkut at the Istanbul Technical University MEM-TEK laboratory. We
thank Mr. Asen Sabuncu (Istanbul Technical University) and Dr. Teach. Assist. Emre
Pınarcı (Çukurova University) for help in drafting the figures. We also thank Assoc. Prof
Dr. Tülay İnan for help in conducting electrical conductivity measurements of the bottled
water samples. This work has been partially supported by Istanbul Technical University
Scientific Research Fund (ITU BAPSIS) Project # 44774.
**Authors contributions**
S.I. and H.C. conceived the project; H.C. collected the samples; N.Y. coordinated
laboratory analysis, compiled seismic events, and prepared the figures; S.I. was the
primary interpreter of the data. S.I. and H.C. were writers of the manuscript with
contributions from N.Y.

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
