# Peer review of "Spring water anomalies before two consecutive earthquakes (Mw 7.7 and Mw 7.6) in Kahramanmaraş (Türkiye) on 6 February 2023"

_Natural Hazards and Earth System Sciences, 2023_

## Referee Comment (RC4)

The work tested the changes in electrical conductivity and the ion concentrations in two springs near the epicenters before and after two earthquakes (Mw 7.7 and Mw 7.6, in Kahramanmaraş), suggesting the anomalies with an increase electrical conductivity and the major ions before the earthquake. The analytical methods used and the results obtained are reliable, but the conclusions are limited due to the limited samples (the number of spring holes) and the significant results from only one point. The manuscript needs major revision. I believe that the authors have done their best to obtain data on all possible spring holes, and it would be difficult to recommend that they test more samples from other possible springs. Therefore, from other perspectives, I think the following possible improvements still exist, depending on the authors.

1. In terms of form, the contents of Table 2 and Figure 3 are somewhat repetitive, and it is suggested that the author could optimize or merge them.
2. Since the author only analyzed the relationship between the ion concentration changes and the earthquake from the perspective of time and location, this analysis is only correlation analysis and lacks causality analysis. Therefore, the author need try to find some data that can reflect the change of crust stress, such as deep drilling data, ground stress station data or satellite observation of surface displacement and deformation data (e.g., InSAR) to support the rationality of ion concentration changes from the perspective of time and space. I think this can greatly increase the reliability of the results of the work.
3. From the time series observation data of the spring water, the increase in ion concentration began as early as one year before the M7.7 earthquake. However, since the author only presented the results of one year, the process of increasing ion concentration was not fully displayed, and the audience could not clearly see when the increase in ion concentration began, and whether it was in a low value stable state more than one year before the M7.7 earthquake. Please add this part of data, if not, please explain the reason.
4. In addition, I would like to know whether the author has obtained synchronized spring temperature data, which I think is also crucial to reflect the process and results of underground fluid migration. If so, it is suggested that the author add relevant content and make analysis. If not, I suggest that the author refer to the schematic diagram in Figure 10 of this paper (He A, Singh R P. Groundwater level response to the Wenchuan earthquake of May 2008[J]. Geomatics, Natural Hazards and Risk, 2018.), and combine the location and lithology of the spring in your work to analyze and discuss the reasons for the abnormal spring.
5. Line 402, two immediate mechanisms was presented. Here the authors should elaborate them in detail as you can as possible, for example providing some schematic diagrams associating to the potential mechanisms. This is very important to understand the physical processes and increase the reliability of this work.

---

## Community Comment (CC2)

[revised manuscript text omitted]

---

## Author Comment (AC3)

**T.C.**
**OSMANİYE VALİLİĞİ**
**İL SAĞLIK MÜDÜRLÜĞÜ**

**KAYNAK SUYU ÜRETİM İZNİ**

| | | | |
|---|---|---|---|
| **Ticari ismi** | : AYRAN SU | **Ruhsat No** | : KS.80.09 |
| **Sahibi** | : Ayran Su Sanayi ve Ticaret A. Ş. | **Ruhsat Tarihi** | : 29.08.2012 |
| **İşleticisi** | : Ayran Su Sanayi ve Ticaret A. Ş. | **Kaptaj Adedi** | : 1 |
| **İşletme Adresi** | : Gökmustafalı Köyü D-400 Karayolu Üzeri 293 Ada 117 Parsel Bahçe İlçesi / OSMANİYE | **Debisi** | : 3.8 L/sn |
| **Kaynak Adresi** | : A.Arıcaklı Köyü Ayranikizharman Mah. Ayran mevkii 376 Ada ve 3 Parsel Bahçe İlçesi / OSMANİYE | **İmla Şekli** | : 19 L. Polikarbonat Damacana, 0.2-0.33-0.4-0.5-1-1.2-1.5-5-10-17-19 L. Pet Şişe, 200-250-300 ml. Pet Bardak. |

Bu Kaynak Suyu Üretim İzni T.C. SAĞLIK BAKANLIĞI tarafından 17.02.2005 tarihli ve 25730 sayılı Resmi Gazete'de yayımlanan "İnsani Tüketim Amaçlı Sular Hakkında Yönetmelik" hükümlerine göre düzenlenmiştir.

**Düzenleme Tarihi:**
- 17.10.2017 tarihinde 17 L. Pet Şişe ilavesi yapılmıştır.

Muammer BALCI
Vali a.
Vali Yardımcısı

**T.C.**
**OSMANİYE VALİLİĞİ**
**İl Sağlık Müdürlüğü**

**KAYNAK SUYU ÜRETİM İZNİNE ESAS ANALİZ SERTİFİKASI**

**Ruhsat No** : KS.80.09

**Tarih** : 29.08.2012
**Düzenleme Tarihi** : 17.10.2017

**KAYNAK SUYUNUN**
**Ticari Adı** : AYRAN SU

**Kaynak Adedi:** 1 (Bir)

**İşletme Adresi:** Gökmustafalı Köyü D-400 Karayolu Üzeri
293 Ada - 117 Parsel Bahçe İlçesi / OSMANİYE

**Kaynak Adresi:** Aşağı Arıcaklı Köyü Ayranikizharman Mah.
Ayran mevkii 376 Ada - 3 Parsel Bahçe İlçesi / OSMANİYE

**Sahibi:** Ayran Su Sanayi ve Ticaret A.Ş.

**Toplam Debisi:** 3.8 L/sn

**İşleticisi:** Ayran Su Sanayi ve Ticaret A.Ş.

**İmla Şekli** : 19 L. Polikarbonat Damacana,
0.2-0.33-0.4-0.5-1-1.2-1.5-5-10-17-19 L. Pet Şişe,
200-250-300 ml.Pet Bardak.

**NİTELİKLERİ**

**A. MİKROBİYOLOJİK PARAMETRELER**

| Parametre | Birim | Değeri |
|---|---|---|
| E. coli | 250 ml' de | 0 |
| Enterokok | 250 ml' de | 0 |
| Koliform bakteri | 250 ml' de | 0 |
| P. aeruginosa | 250 ml' de | 0 |
| Anaerob sporlu sülfit redükte eden bakteriler | 50 ml' de | 0 |
| Patojen stafilokoklar | 100 ml' de | 0 |
| Koloni sayısı (22 °C) | ml' de | 0 |
| Koloni sayısı (37 °C) | ml' de | 0 |
| Parazitler | 100 ml' de | 0 |

**B. KİMYASAL PARAMETRELER**

| | | |
|---|---|---|
| Akrilamid | µg/L | 0 |
| Antimon | µg/L | 0 |
| Arsenik | µg/L | 0 |
| Benzen | µg/L | 0 |
| Benzo(a)piren | µg/L | 0 |
| Bor | mg/L | 0 |
| Bromat | µg/L | 0 |
| Kadmiyum | µg/L | 0 |
| Krom | µg/L | 0 |
| Bakır | mg/L | 0 |
| Siyanür | µg/L | 0 |
| 1,2-dikloretan | µg/L | 0 |
| Epikloridin | µg/L | 0 |
| Florür | mg/L | 0 |
| Kurşun | µg/L | 0 |
| Civa | µg/L | 0 |
| Nikel | µg/L | 0 |
| Nitrat | mg/L | 5.59 |
| Nitrit | mg/L | 0 |

**B. KİMYASAL PARAMETRELER (devam)**

| | | |
|---|---|---|
| Pestisitler | µg/L | 0 |
| Toplam pestisitler | µg/L | 0 |
| PAH (Polisiklik aromatik hidrokarbonlar) | µg/L | 0 |
| Selenyum | µg/L | 0 |
| Tetrakloreten ve trikloreten | µg/L | 0 |
| Toplam Trihalometanlar | µg/L | 0 |
| Vinil Klorür | µg/L | 0 |

**C. GÖSTERGE PARAMETRELERİ**

| | | |
|---|---|---|
| Koloni sayısı (22 °C) | ml' de | 0 |
| Koliform bakteri | 100 ml' de | 0 |
| Renk | | Uygun |
| Koku | | Uygun |
| Tat | | Uygun |
| Bulanıklık | | Uygun |
| pH | pH birimleri | 7.44 |
| İletkenlik | 20 °C'de µS/cm | 110 |
| Alüminyum | µg/L | 0 |
| Amonyum | mg/L | 0 |
| Klorür | mg/L | 3.12 |
| Demir | µg/L | 0 |
| Mangan | µg/L | 0 |
| Oksitlenebilirlik | mg/L $O_2$ | 0 |
| Sülfat | mg/L | 8.37 |
| Sodyum | mg/L | 3.86 |
| TOC (Toplam Organik Karbon) | µg/L | 0 |

**D. RADYOAKTİVİTE**

| | | |
|---|---|---|
| Trityum | Bq/L | Uygun |
| Toplam gösterge dozu | mSv/yıl | Uygun |

Muammer BALCI
Vali a.
Vali Yardımcısı

[Figure]

**ANALİZ RAPORU**

| | | |
|---|---|---|
| Alüminyum (Al) | 0 | µg/L |
| Amonyum (NH₄) | 0 | mg/L |
| Klorür (Cl) | 3.12 | mg/L |
| İletkenlik | 110 | 20°C'de µS/cm |
| pH | 7.44 | pH Birimleri |
| Demir (Fe) | 0 | µg/L |
| Mangan (Mn) | 0 | µg/L |
| Oksitlenebilirlik | 0 | mg/L O₂ |
| Sülfat (SO₄) | 8.37 | mg/L |
| Sodyum (Na) | 3.86 | mg/L |
| Renk | Uygun | - |
| Koku | Uygun | - |
| Tat | Uygun | - |
| Bulanıklık | Uygun | - |
| Koliform Bakteri | 0 | kob/250 mL |
| 22°C'de Koloni Sayısı | 0 | kob/mL |

T.C. Osmaniye Valiliği'nin 29.08.2012 tarih ve KS.80.08 sayılı üretim izni ile Ayran Su San. ve A.Ş. tesislerinde el değmeden tam otomatik makinalarda doldurulmuştur.

Üretim, son kullanma tarihi parti ve seri numaraları şişenin üzerindedir. Üretim Sazlıpen numarası yerine geçer.

Tekniğine uygun olarak ozonla zenginleştirilmiş havsı ile oksijenleme işlemine tabi tutulmuştur.

Ayran Su San. ve Tic. A.Ş.
D400 Karayolu Üzeri
Gökmustafalı Köyü
Bahçe - Osmaniye - TÜRKİYE
Tel : 0 328 400 00 80
Fax : 0 328 600 00 81

ISO 22000 - ISO 18000 - ISO 9001
info@ayransu.com.tr
www.ayransu.com.tr

**19L**

BAHÇE

DOĞAL KAYNAK SUYU

8 680596 730028

T.C.
SAĞLIK BAKANLIĞI
OSMANİYE İL SAĞLIK MÜDÜRLÜĞÜ 8.01.2022

Süleyman DOĞAN
Uzman
Çevre Sağlığı Birimi

Uzm.Dr. Ersin PEKER
Halk Sağlığı Hizmetleri
Başkan Yardımcısı

Dr. Mehmet BAHAR
Halk Sağlığı Hizmetleri
Başkanı

Dr. Hasan ÖZNAVRUZ
İl Sağlık Müdürü

AYRAN SU SANAYİ VE TİC. A.Ş.
Osmaniye V.D. V. No: 12 45
Osmaniye Ticaret Sicil No: B.990
D-400 Karayolu Üzeri Gökmustafalı Köyu
Tel 0 328 600 0081
BAHÇE OSMANİYE

[Figure]

**ANALİZ RAPORU**

| | | |
|---|---|---|
| Alüminyum (Al) | 0 | µg/L |
| Amonyum (NH₄) | 0 | mg/L |
| Klorür (Cl) | 3.12 | mg/L |
| İletkenlik | 110 | 20°C'de µS/cm |
| pH | 7.44 | pH Birimleri |
| Demir (Fe) | 0 | µg/L |
| Mangan (Mn) | 0 | µg/L |
| Oksitlenebilirlik | 0 | mg/L O₂ |
| Sülfat (SO₄) | 8.37 | mg/L |
| Sodyum (Na) | 3.86 | mg/L |
| Renk | Uygun | - |
| Koku | Uygun | - |
| Tat | Uygun | - |
| Bulanıklık | Uygun | - |
| Koliform Bakteri | 0 | kob/250 mL |
| 22°C'de Koloni Sayısı | 0 | kob/mL |

T.C. Osmaniye Valiliği'nin 29.06.2012 tarih ve KS.60.09 sayılı üretim izni ile Ayran Su San. ve A.Ş. tesislerinde el değmeden tam otomatik makinalarda doldurulmuştur.

Üretim, son kullanma tarihi parti ve seri numaraları şişenin üzerindedir.
Üretim Sıazfı,seri numarası yerine geçer.

Tekniğine uygun olarak ozonla zenginleştirilmiş hava ile oksijenlenme işlemine tabi tutulmuştur.

Ayran Su San. ve Tic. A.Ş.
D400 Karayolu Üzeri
Gökmustafalı Köyü
Bahçe - Osmaniye - TÜRKİYE
Tel : 0 328 600 00 81
Fax : 0 328 600 00 81

ISO 22000 - ISO 18000 - ISO 9001
info@ayransu.com.tr
www.ayransu.com.tr

**ayran Su**
DOĞAL KAYNAK SUYU

BAHÇE

**19L**
GERİ DÖNÜŞÜMSÜZ PET

Süleyman DOĞAN
Uzman
Çevre Sağlığı Birimi

Uzm.Dr. Ersin PEKER
Halk Sağlığı Hizmetleri
Başkan Yardımcısı

Dr. Mehmet BAHAR
Halk Sağlığı Hizmetleri
Başkanı

Dr. Hasan ÖZNAVRUZ
İl Sağlık Müdürü

T.C.
SAĞLIK BAKANLIĞI
OSMANİYE İL SAĞLIK MÜDÜRLÜĞÜ

18.01.2022

[Figure]

AYRAN SU SANAYİ VE TİC. A.Ş.
Osmaniye V.D. T.No: 104345
Osmaniye Tic. Sicil No: 8.990
D-400 Karayolu Mustafalı Köyü
Tel: 0 328 ... 08 ... 500 0081
BAHÇE OSMANİYE

[Figure]

**ANALİZ RAPORU**

| | | |
|---|---|---|
| Alüminyum (Al) | 0 | µg/L |
| Amonyum (NH₄) | 0 | mg/L |
| Klorür (Cl) | 3.12 | mg/L |
| İletkenlik | 110 | 20°C'de µS/cm |
| pH | 7.44 | pH Birimleri |
| Demir (Fe) | 0 | µg/L |
| Mangan (Mn) | 0 | µg/L |
| Oksitlenebilirlik | 0 | mg/L O₂ |
| Sülfat (SO₄) | 8.37 | mg/L |
| Sodyum (Na) | 3.86 | mg/L |
| Renk | Uygun | - |
| Koku | Uygun | - |
| Tat | Uygun | - |
| Bulanıklık | Uygun | - |
| Koliform Bakteri | 0 | kob/250 mL |
| 22°C'de Koloni Sayısı | 0 | kob/mL |

T.C. Osmaniye Valiliği'nin 20.08.2012 tarih ve KS.80.09 sayılı üretim izni ile Ayran Su San. ve A.Ş. tesislerinde el değmeden tam otomatik makinalarda doldurulmuştur.

Üretim, son kullanma tarihi parti ve seri numaraları şişenin üzerindedir.
Üretim Sızıntıları numaraları yerine geçer.

Tekniğine uygun olarak sporla zenginleştirilmiş hava ile oksijenleme işlemine tabi tutulmuştur.

Ayran Su San. ve Tic. A.Ş.
D400 Karayolu Üzeri
Gökmustafalı Köyü
Bahçe - Osmaniye - TÜRKİYE
Tel : 0 328 600 00 80
Fax: 0 328 600 00 81

ISO 22000 - ISO 18000 - ISO 9001
info@ayransu.com.tr
www.ayransu.com.tr

**17L**
GERİ DÖNÜŞÜMSÜZ PET

BAHÇE

**ayran Su**
DOĞAL KAYNAK SUYU

8 680596 739687

18.01.2022

T.C.
SAĞLIK BAKANLIĞI
OSMANIYE İL SAĞLIK MÜDÜRLÜĞÜ

Süleyman DOĞAN
Uzman
Çevre Sağlığı Birimı

Uzm.Dr. Ersin PEKER
Halk Sağlığı Hizmetleri
Başkan Yardımcısı

Dr. Mehmet BAHAR
Halk Sağlığı Hizmetleri
Başkanı

Dr. Hasan ÖZNAVRUZ
İl Sağlık Müdürü

AYRAN SU SANAYİ VE TİC. A.Ş.
Osmaniye V.D. No: 122104345
Osmaniye Tic. Sic. No: B.990
D400 Karayolu Üzeri Gökmustafalı Köyü
Tel: 0 328 600 0081
BAHÇE - OSMANIYE

[Figure]

**ANALİZ RAPORU**

| | | |
|---|---|---|
| Alüminyum (Al) | 0 | µg/L |
| Amonyum (NH₄) | 0 | mg/L |
| Klorür (Cl) | 3.12 | mg/L |
| İletkenlik | 110 | 20°C'de µS/cm |
| pH | 7.44 | pH Birimleri |
| Demir (Fe) | 0 | µg/L |
| Mangan (Mn) | 0 | µg/L |
| Oksitlenebilirlik | 0 | mg/L O₂ |
| Sülfat (SO₄) | 8.37 | mg/L |
| Sodyum (Na) | 3.86 | mg/L |
| Renk | Uygun | - |
| Koku | Uygun | - |
| Tat | Uygun | - |
| Bulanıklık | Uygun | - |
| Koliform Bakteri | 0 | kob/250 mL |
| 22°C'de Koloni Sayısı | 0 | kob/mL |

T.C. Osmaniye Valiliği'nin 29.08.2012 tarih ve KS.80.09 sayılı üretim izni ile Ayran Su San. ve A.Ş. tesislerinde el değmeden tam otomatik makinalarda doldurulmuştur.

Üretim, son kullanma tarihi parti ve seri numarası geçerli üzerindedir.

Üretim Saat/saat numarası yerine geçer.

Ayran Su San. ve Tic. A.Ş.
D400 Karayolu Üzeri
Gökmustafalı Köyü
Bahçe - Osmaniye - TÜRKİYE
Tel.: 0.328 600 00 80
Fax.: 0.328 600 00 81

**ayran su**
DOĞAL KAYNAK SUYU

BAHÇE

5L

Süleyman DOĞAN
Uzman
Çevre Sağlığı Birimi

Uzm.Dr. Ersin PEKER
Halk Sağlığı Hizmetleri
Başkan Yardımcısı

Dr. Mehmet BAHAR
Halk Sağlığı Hizmetleri
Başkanı

Dr. Hasan ÖZNAVRUZ
İl Sağlık Müdürü

T.C.
SAĞLIK BAKANLIĞI
OSMANİYE İL SAĞLIK MÜDÜRLÜĞÜ

18.07.2022

AYRAN SU SANAYİ VE TİC. A.Ş.
Osmaniye V... No: ...84345
Osmaniye Tic. Odası Sicil No: 8.990
D-400 Kara... Üzeri ...kmustafalı Köyü
Tel 0.3... ...00 ...8 600 0081
BAHÇE OSMANİYE

[Figure]

**BAHÇE**

**ayran** *Su*
DOĞAL KAYNAK SUYU

1,5 L

**ANALİZ RAPORU**

| | | | | | |
|---|---|---|---|---|---|
| Alüminyum (Al) | 0 | µg/L | Oksitlenebilirlik | 0 | mg/L O₂ |
| Amonyum (NH₄) | 0 | mg/L | Sülfat (SO₄) | 8.37 | mg/L |
| Klorür (Cl) | 3.12 | mg/L | Sodyum (Na) | 3.86 | mg/L |
| İletkenlik | 110 | 20°C'de µS/cm | Renk | Uygun | - |
| pH | 7.44 | pH Birimi | Koku | Uygun | - |
| Demir (Fe) | 0 | µg/L | Tat | Uygun | - |
| Mangan (Mn) | 0 | µg/L | Bulanıklık | Uygun | - |

T.C. Osmaniye Valiliği'nin 29.08.2012 tarih ve KS.80.09 sayılı
üretim izni ile Ayran Su San. ve Tic. A.Ş. Tesisleriade el değmeden
tam otomatik makinalarda doldurulmuştur.
Temiz, kuru, serin ve kokusuz bir ortamda, güneş
ışığından koruyarak saklayınız.

Üretim, son kullanma tarihi parti ve seri numaraları şişenin üzerindedir.
Üretim saati seri numarası yerine geçer.

Kokuşlığını aşırı sıcakta zararlı yangırlıkdilirhş beısa ile oksijenleme
gözmsu tür alanılandar
ISO 9001 ISO 22000 ISO 18000
Ayran Su San. ve Tic. A.Ş.
D.400 Karayolu Üzeri Gökmustafalı Köyu   Bahçe / Osmaniye / Türkiye
Tel: 0.328. 600 00 80   Fax: 0.328. 600 00 81
www.ayransu.com.tr

**ayran** *Su*
DOĞAL KAYNAK SUYU

1,5 L

8 680596 730042

FSC 266

T.C.
SAĞLIK BAKANLIĞI
OSMANİYE İL SAĞLIK MÜDÜRLÜĞÜ   18.01.2022

Süleyman DOĞAN
Uzman
Çevre Sağlığı Birimı

Uzm.Dr. Ersin PEKER
Halk Sağlığı Hizmetleri
Başkan Yardımcısı

Dr. Mehmet BAHAR
Halk Sağlığı Hizmetleri
Başkanı

Dr. Hasan ÖZNAVRUZ
İl Sağlık Müdürü

AYRAN SU SANAYİ VE TİC. A.Ş.
Osmaniye V.D. V.No: 1230704345
Osmaniye Tic. Mrsz. Sicil No: 8.990
D-400 Karayolu Üzeri Gökmustafalı Köyü
Tel: 0 328 600 0081     600 0081
BAHÇE   OSMANİYE

[Figure]

**ANALİZ RAPORU**

| | | | | | | |
|---|---|---|---|---|---|---|
| Alüminyum (Al) | 0 | µg/L | Oksitlenebilirlik | 0 | mg/L O₂ | |
| Amonyum (NH4) | 0 | mg/L | Sülfat (SO4) | 6.37 | mg/L | |
| Klorür (Cl) | 3.12 | mg/L | Sodyum (Na) | 3.86 | mg/L | |
| İletkenlik | 110 | 20°C'de µS/cm | Renk | Uygun | | - |
| pH | 7.44 | pH Birimi | Koku | Uygun | | - |
| Demir (Fe) | 0 | µg/L | Tat | Uygun | | - |
| Mangan (Mn) | 0 | µg/L | Bulanıklık | Uygun | | - |

T.C. Osmaniye Valiliği'nin 29.08.2012 tarih ve KS.80.09 sayılı
üretim izni ile Ayran Su San. ve Tic. A.Ş. Tesislerinde el değmeden
tam otomatik makinalarda doldurulmuştur.

Temiz, kuru, serin ve kokusuz bir ortamda, güneş
ışığından koruyarak saklayınız.

Üretim, son kullanma tarihi parti ve seri numaraları şişenin üzerindedir.
Üretim saati seri numarası yerine geçer.

Teknolojisi, uygun olarak ozonla zenginleştirilmiş havai ile oksijenleme
işlemine tabi tutulmuştur.

ISO 9001 ISO 22000 ISO 18000
Ayran Su San. ve Tic. A.Ş.
D.400 Korayolu Üzeri Gökmustafalı Köyü   Bahçe / Osmaniye / Türkiye
Tel: 0.328. 600 00 80. Fax: 0.328. 600 00 81
www.ayransu.com.tr

BAHÇE

ayran Su
DOĞAL KAYNAK SUYU
1,2 L

ayran Su
DOĞAL KAYNAK SUYU
1,2 L

8 680596 739922

Süleyman DOĞAN
Uzman
Çevre Sağlığı Birimi

Uzm.Dr. Ersin PEKER
Halk Sağlığı Hizmetleri
Başkan Yardımcısı

Dr. Mehmet BAHAR
Halk Sağlığı Hizmetleri
Başkanı

Dr. Hasan ÖZNAVRUZ
İl Sağlık Müdürü

18.01.2022

T.C.
SAĞLIK BAKANLIĞI
OSMANİYE İL SAĞLIK MÜDÜRLÜĞÜ

AYRAN SU SANAYİ VE TİC. A.Ş.
Osmaniye V.D. V.No: 127 104345
Osmaniye Ticaret Odası Sicil no: 8.990
D-400 Karayolu Üzeri Gökmustafalı Köyü
Tel: 0 328 600 0080    328 600 0081
BAHÇE   OSMANİYE

[Figure]

Süleyman DOĞAN
Uzman
Çevre Sağlığı Birimi

Uzm.Dr. Ersin PEKER
Halk Sağlığı Hizmetleri
Başkan Yardımcısı

Dr. Mehmet BAHAR
Halk Sağlığı Hizmetleri
Başkanı

T.C.
SAĞLIK BAKANLIĞI
OSMANİYE İL SAĞLIK MÜDÜRLÜĞÜ     18.01.2022

Dr. Hasan ÖZNAVRUZ
İl Sağlık Müdürü

AYRAN SU SANAYİ VE TİC. A.Ş.
Osmaniye V.D. V.No: 130104345
Osmaniye Tic. Odası Sicil No: B 990
D-400 Karayolu üzeri Gökmustafalı Köyü
Tel: 0 328        600 0081
BAHÇE   OSMANİYE

[Figure]

**ANALİZ RAPORU**

| | | | | | |
|---|---|---|---|---|---|
| Alüminyum (Al) | 0 | µg/L | Oksidenebilirlik | 0 | mg/L O₂ |
| Amonyum (NH₄) | 0 | mg/L | Sülfat (SO₄) | 8.37 | mg/L |
| Klorür (Cl) | 3.12 | mg/L | Sodyum (Na) | 3.86 | mg/L |
| İletkenlik | 110 | 20°C'de µS/cm | Renk | Uygun | - |
| pH | 7.44 | pH Birimi | Koku | Uygun | - |
| Demir (Fe) | 0 | µg/L | Tat | Uygun | - |
| Mangan (Mn) | 0 | µg/L | Bulanıklık | Uygun | - |

T.C. Osmaniye Valiliği'nin 29.08.2012 tarih ve KS.80.09 sayılı
üretim izni ile Ayran Su San. ve Tic. A.Ş. Tesislerinde el değmeden
tam otomatik makinalarda doldurulmuştur.

**Temiz, kuru, serin ve kokusuz bir ortamda, güneş
ışığından koruyarak saklayınız.**

Üretim, son kullanma tarihi parti ve seri numaraları şişenin üzerindedir.
Üretim saati seri numarası yerine geçer.

Tekrağı önemli: uygun olmayan bir ortamda sağlığınıza tehlike ile karşılaşma
olabilir.

İSO 9001 İSO 22000 İSO 18000
Ayran Su San. ve Tic. A.Ş.
D-400 Karayolu Üzeri Gökmustafalı Köyü   Bahçe / Osmaniye / Türkiye
Tel: 0.328. 600 00 80 Fax: 0.328. 600 00 81
www.ayransu.com.tr

**DOĞAL KAYNAK SUYU**

**0,5 L**

8 680596 730035

Süleyman DOĞAN
Uzman
Çevre Sağlığı Birimi

**Uzm.Dr. Ersin PEKER**
Halk Sağlığı Hizmetleri
Başkan Yardımcısı

**Dr. Mehmet BAHAR**
Halk Sağlığı Hizmetleri
Başkanı

Dr. Hasan ÖZNAVRUZ
İl Sağlık Müdürü

**AYRAN SU SANAYİ VE TİC. A.Ş.**
Osmaniye V.D. V.Nr. 1342-94345
Osmaniye Ticaret Sicil No. 8.990
D-400 Karayolu Üzeri Gökmustafalı Köyu
Tel: 0 328        600 0081
BAHÇE   OSMANİYE

T.C.
SAĞLIK BAKANLIĞI
OSMANİYE İL SAĞLIK MÜDÜRLÜĞÜ

18.01.2022

[Figure]

**ANALİZ RAPORU**

| | | | | | |
|---|---|---|---|---|---|
| Alüminyum (Al) | 0 | µg/L | Oksitlenebilirlik | 0 | mg/L O₂ |
| Amonyum (NH₄) | 0 | mg/L | Sülfat (SO₄) | 6.37 | mg/L |
| Klorür (Cl) | 3.12 | mg/L | Sodyum (Na) | 3.85 | mg/L |
| İletkenlik | 110 | 20°C'de µS/cm | Renk | Uygun | - |
| pH | 7.44 | pH Birimleri | Koku | Uygun | - |
| Demir (Fe) | 0 | µg/L | Tat | Uygun | - |
| Mangan (Mn) | 0 | µg/L | Bulanıklık | Uygun | - |

T.C. Osmaniye Valiliği'nin 29.08.2012 tarih ve KS.80.09 sayılı
üretim izni ile Ayran San. ve Tic. A.Ş. Tesislerinde el değmeden
tam otomatik makinalarda doldurulmuştur.
Temiz, kuru, serin ve kokusuz bir ortamda, güneş
ışığından koruyarak saklayınız.

Üretim, son kullanma tarihi parti ve seri numarası şişenin üzerindedir.

ISO 9001 ISO 22000 ISO 18000
Ayran Su San. ve Tic. A.Ş.
D.400 Karayolu Üzeri Gökmustafalı Köyü    Bahçe / Osmaniye / Türkiye
Tel: 0.328. 600 00 80  Fax: 0.328. 600 00 81
www.ayransu.com.tr

BAHÇE

**ayran Su**

DOĞAL KAYNAK SUYU

0,4 L

**ayran Su**

DOĞAL KAYNAK SUYU

0,4 L

8 680596 739939

18.01.2022

T.C.
SAĞLIK BAKANLIĞI
OSMANİYE İL SAĞLIK MÜDÜRLÜĞÜ

Süleyman DOĞAN
Uzman
Çevre Sağlığı Birimi

Uzm.Dr. Ersin PEKER
Halk Sağlığı Hizmetleri
Başkan Yardımcısı

Dr. Mehmet BAHAR
Halk Sağlığı Hizmetleri
Başkanı

Dr. Hasan ÖZNAVRUZ
İl Sağlık Müdürü

AYRAN SU SANAYİ VE TİC. A.Ş.
Osmaniye V.D. V.No: 1230104345
Osmaniye Tic. Odası Sicil No: B.990
D-400 Karayolu Üzeri Gökmustafalı Köyü
Tel: 0 328 6     600 0081
BAHÇE   OSMANİYE

[Figure]

**ANALİZ RAPORU**

| | | | | | |
|---|---|---|---|---|---|
| Alüminyum (Al) | 0 | µg/L | Oksitlenebilirlik | 0 | mg/L O₂ |
| Amonyum (NH₄) | 0 | mg/L | Sülfat (SO₄) | 8.37 | mg/L |
| Klorür (Cl) | 3.12 | mg/L | Sodyum (Na) | 3.86 | mg/L |
| İletkenlik | 110 | 20°C'de µS/cm | Renk | Uygun | - |
| pH | 7.44 | pH Birimleri | Koku | Uygun | - |
| Demir (Fe) | 0 | µg/L | Tat | Uygun | - |
| Mangan (Mn) | 0 | µg/L | Bulanıklık | Uygun | - |

T.C. Osmaniye Valiliği'nin 29.08.2012 tarih ve KS.80.09 sayılı üretim izni ile Ayran Su San. ve Tic. A.Ş. Tesislerinde el değmeden tam otomatik makinalarda doldurulmuştur.

Temiz, kuru, serin ve kokusuz bir ortamda, güneş ışığından koruyarak saklayınız.

Üretim, son kullanma tarihi parti ve seri numaraları şişenin üzerindedir.

İçtiğiniz suyun düzenli analiz raporlarını kıssa ile oksijenleme idrarını tali faaliyetdedir.

ISO 9001 ISO 22000 ISO 18000
Ayran Su San. ve Tic. A.Ş.
D-400 Karayolu Üzeri Gökmustafalı Köyü · Bahçe / Osmaniye / Türkiye
Tel: 0.328. 600 00 80 Fax: 0.328. 600 00 81
www.ayransu.com.tr

**BAHÇE**

**ayran** Su
DOĞAL KAYNAK SUYU
0,33 L

**ayran** Su
DOĞAL KAYNAK SUYU
0,33 L

8 680596 730066

Süleyman DOĞAN
Uzman
Çevre Sağlığı Birimi

Uzm.Dr. Ersin PEKER
Halk Sağlığı Hizmetleri
Başkan Yardımcısı

Dr. Mehmet BAKAR
Halk Sağlığı Hizmetleri
Başkanı

Dr. Hasan ÖZNAVRUZ
İl Sağlık Müdürü

T.C.
SAĞLIK BAKANLIĞI
OSMANİYE İL SAĞLIK MÜDÜRLÜĞÜ 08.01.2022

AYRAN SU SANAYİ VE TİC. A.Ş.
Osmaniye V.D. V.No: 1230104345
Osmaniye Tic. Odası Sicil No: 5.990
D-400 Karayolu Üzeri Gökmustafalı Köyü
Tel: 0.328. 600 0081
BAHÇE OSMANİYE

[Figure]

**ANALİZ RAPORU**

| | | | | | |
|---|---|---|---|---|---|
| Alüminyum (Al) | 0 | µg/L | Oksitlenebilirlik | 0 | mg/L O₂ |
| Amonyum (NH₄) | 0 | mg/L | Sülfat (SO₄) | 6.37 | mg/L |
| Klorür (Cl) | 3.12 | mg/L | Sodyum (Na) | 3.86 | mg/L |
| İletkenlik | 110 | 20°C'de µS/cm | Renk | Uygun | |
| pH | 7.44 | pH Birimleri | Koku | Uygun | |
| Demir (Fe) | 0 | µg/L | Tat | Uygun | |
| Mangan (Mn) | 0 | µg/L | Bulanıklık | Uygun | |

T.C. Osmaniye Valiliği'nin 29.08.2012 tarih ve KS.80.09 sayılı üretim izni ile Ayran Su San. ve Tic. A.Ş. Tesislerinde el değmeden tam otomatik makinalarda doldurulmuştur.
Temiz, kuru, serin ve kokusuz bir ortamda, güneş ışığından koruyarak saklayınız.

Üretim, son kullanma tarihi parti ve seri numaraları şişenin üzerindedir. Üretim saati seri numarası yerine geçer.

Tekniğine uygun olarak mutfak eşyalarıdezenfekte deva ile oksijenleme olacsmon için tesleolunnr.

ISO 9001 ISO 22000 ISO 18000
Ayran Su San. ve Tic. A.Ş.
D-400 Karayolu Üzeri Gökmustafalı Köyü   Bahçe / Osmaniye / Türkiye
Tel: 0.328. 600 00 80 Fax: 0.328. 600 00 81
www.ayransu.com.tr

**BAHÇE**

**ayran** Su
DOĞAL KAYNAK SUYU
0,2 L

**ayran** Su
DOĞAL KAYNAK SUYU
0,2 L

8 680596 739670

T.C.
SAĞLIK BAKANLIĞI
OSMANİYE İL SAĞLIK MÜDÜRLÜĞÜ

T.C.
SAĞLIK BAKANLIĞI
OSMANİYE İL SAĞLIK MÜDÜRLÜĞÜ
1923

18.01.2022

Süleyman DOĞAN
Uzman
Çevre Sağlığı Birimi

Uzm.Dr. Ersin PEKER
Halk Sağlığı Hizmetleri
Başkan Yardımcısı

Dr. Mehmet BAHAR
Halk Sağlığı Hizmetleri
Başkanı

Dr. Hasan ÖZNAVRUZ
İl Sağlık Müdürü

AYRAN SU SANAYİ VE TİC. A.Ş.
Osmaniye V.D. V.No: 122 04345
Osmaniye Tic. Sic. No: 8.990
D-400 Karayolu Üzeri Gökmustafalı Köyü
Tel: 0 328 600 081   600 0081
BAHÇE   OSMANİYE

[Figure]

**ANALİZ RAPORU**

| | | | | | | |
|---|---|---|---|---|---|---|
| Alüminyum (Al) | 0 | µg/L | Oksitlenebilirlik | 0 | mg/L $O_2$ |
| Amonyum ($NH_4$) | 0 | mg/L | Sülfat ($SO_4$) | 8.37 | mg/L |
| Klorür (Cl) | 3.12 | mg/L | Sodyum (Na) | 3.86 | mg/L |
| İletkenlik | 110 | 20°C'de µS/cm | Renk | Uygun | |
| pH | 7.44 | | Koku | Uygun | |
| Demir (Fe) | 0 | µg/L | Tat | Uygun | |
| Mangan (Mn) | 0 | µg/L | Bulanıklık | Uygun | |

T.C. Osmaniye Valiliği'nin 29.08.2012 tarih ve KS.80.09 Sayılı üretim izni ile
Ayran Su San. ve Tic. A.Ş. Tesislerinde el değmeden tam otomatik makinalarda doldurulmuştur.

Temiz, kuru, serin ve kokusuz bir ortamda güneş ışığından koruyarak saklayınız.

ISO 9001 ISO 22000 ISO 18000          www.ayransu.com.tr

Ayran Su San. ve Tic. A.Ş.
D400 Karayolu Üzeri
Göksmestafalı Köyü
Bahçe / Osmaniye / Türkiye
Tel: 0328 600 00 80
Fax: 0328 600 00 81

PSE 266

Üretim, son tüketim tarihi, parti ve seri numaraları, dolum hacmi bardağın altındadır. Üretim tarihi, parti numarası yerine, Üretim saati seri numarası yerine geçer.

Tekniğine uygun olarak, ozonla zenginleştirilmiş hava ile oksijenleme işlemine tabi tutulmuştur.

**DOĞAL KAYNAK SUYU**

YILPAR AMBALAJ
İŞLETME KAYIT NO TR-35.4-025021

T.C.
SAĞLIK BAKANLIĞI
OSMANİYE İL SAĞLIK MÜDÜRLÜĞÜ

18.07.2022

Süleyman DOĞAN
Uzman
Çevre Sağlığı Birimi

Uzm.Dr. Ersin PEKER
Halk Sağlığı Hizmetleri
Başkan Yardımcısı

Dr. Mehmet BAHAR
Halk Sağlığı Hizmetleri
Başkanı

Dr. Hasan ÖZNAVRUZ
İl Sağlık Müdürü

AYRAN SU SANAYİ VE TİC A.Ş.
Osmaniye V.D. V.No
Osmaniye Tic. Odası
D-400 Karayolu
Tel 0 32
600 0081
BAHÇE   OSMANİYE

---

## Author Response (AR1)

**RESPONSE LETTER**

4 December 2023

Prof. Dr. Filippos Vallianatos, NHESS Editor,

Dear Prof. Vallianatos,

Thank you for kind consideration and prompt handling of our manuscript # nhess-2023-133. Thanks for the kind decision as "minor revision" on our manuscript.

We thank all **five referees**, Dr. Giovanni Martinelli, Dr. Vivek Walia, and three anonymous referees, for constructive reviews and discussion they have posted on discussion forum on the NHESS website. Most of our responses were already posted in the discussion forum on the NHESS website.

We are pleased to inform you that we benefitted greatly from the referees' suggestions and comments in revising the manuscript.

We submit our revised manuscript for your kind evaluation and publication.

Prof. Dr. Sedat Inan

Corresponding author

**Below we provide our responses (in blue fonts) to each referees' comments/suggestions (black fonts).**

**Referee 1: Dr. Giovanni Martinelli**

**Comment 1:**

I have found interesting the manuscript https://doi.org/10.5194/nhess-2023-133 submitted by Inan et al. In particular Authors reported that commercially packaged spring water samples dated prior to and subsequent to the seismic events occurring on February 6,2023 were analyzed. Hydrogeochemical precursors have been identified in bottled natural spring waters originating from Ayran Spring and Bahçepınar Spring. These springs are located around 100 km and 175 km away from the epicentres of the Mw 7.7 and Mw 7.6 Kahramanmaraş (Türkiye) Earthquakes that occurred on 6 February 2023, respectively. The water samples at hand encompass the temporal span from March 2022 to March 2023. The pre-earthquake anomaly is distinguished by an elevation in electrical conductivity and the presence of significant ions (such as $Ca^{2+}$, $Mg^{2+}$, $K^+$, $Na^+$, $Cl^-$, and $SO_4^{2-}$) in Ayran Spring water samples, as opposed to the baseline conditions. The concentration of main ions exhibited a significant decrease, and the levels of ions gradually approached the background values approximately fourteen days following the occurrence of the earthquakes. The Bahçepınar water is obtained from shallow boreholes that have been excavated in alluvial deposits. These deposits are believed to be disconnected from the underlying basement rocks, which could explain the absence of anomalous water chemistry prior to the occurrence of earthquakes. This observation substantiates the significance of sampling locations in the identification of potential indicators of earthquakes. The findings pertaining to the Ayran spring water samples suggest that the chemical anomalies observed in discrete samples of spring water can offer significant insights about the pre-earthquake deformation processes of the Earth's crust. The implementation of long-term monitoring techniques, such as the monitoring of spring waters, within a multidisciplinary network, holds the potential to yield credible proxy indicators of pre-earthquake crustal deformation. Continuous earthquake monitoring networks of geophysical and geochemical parameters should be established and run for a sufficiently long time. The paper is well organized and I hope i twill be soon published after the addition of further References in which geochemical variations have been observed in geofluids (in general) in Turkey.

We are glad that you have liked the manuscript and found it suitable for publication in the journal. Thank you Dr. Martinelli for your kind words about our manuscript. We will add further relevant references as you suggest.

Furthermore, Inan et al (2010) and Inan et al (2012c) as cited in the manuscript mention hydrogeochemical anomalies before Mw 4.2 and Mw 4.8 earthquakes in Aegean Extensional Province and the 23 October Mw 7.2 Van earthquake.

Furthermore, we have added the below the relevant six references in the revised manuscript.

- Vallianatos, F., Triantis, D., Tzanis, A., Anastasiadis, C., Stavrakas, I., 2004. Electric earthquake precursors: from laboratory results to field observations. Physics and Chemistry of the Earth Parts A/B/C 29, 339-351. https://doi.org/10.1016/j.pce.2003.12.003

- Vallianatos, F. and Tzanis, A., 1998. Electric current generation associated with the deformation rate of a solid: Preseismic and coseismic signals.  Physics and Chemistry of the Earth Parts A/B/C 23, 933-938**.** https://doi.org/10.1016/S0079-1946(98)00122-0
- Rapti, D., Martinelli, G., Zheng, G., Vincenzi, C., 2023. Bottled Mineral Waters as Unconventional Sampling in Hydro-Geological Research. Water 15, 3466. https://doi.org/10.3390/w15193466
- Martinelli, G. and Ferrari, G., 1991. Earthquake forerunners in a selected area of Northern Italy: recent developments in automatic geochemical monitoring. Tectonophysics 193, 397-410. https://doi.org/10.1016/0040-1951(91)90348-V
- Walia, V., Yang, T.F., Hong, W.L., Lin, S.J., Fu, C.C., Wen, K.L., Chen, C.H. 2009. Geochemical variation of soil–gas composition for fault trace and earthquake precursory studies along the Hsincheng fault in NW Taiwan, Applied Radiation and Isotopes 67, 1855-1863, https://doi.org/10.1016/j.apradiso.2009.07.004
- He, A. and Singh, R.P., 2019. Groundwater level response to the Wenchuan earthquake of May 2008, Geomatics, Natural Hazards and Risk 10, 336-352, https://doi.org/10.1080/19475705.2018.1523236

**Comment 2: RC2**: 'Reply on AC1', Giovanni Martinelli, 16 Aug 2023
I hope the paper will be quickly accepted and published after the accepted addition.

Thank you Dr. Martinelli for your support.

**Referee 2: Anonymous**

Below are **our responses in blue font** to the **referee's remarks in black fonts.**

The manuscript "Spring water anomalies before two consecutive earthquakes (Mw 7.7 and Mw 7.6) in Kahramanmaraş (Türkiye) on 6 February 2023," by Inan and colleagues, reports the description of geochemical anomalies in a spring before the 2 earthquakes on February 6, 2023. The article provides a comprehensive overview of the current state of knowledge, detailing the methodology employed and presenting the obtained results effectively. I think that publishing the observations made in this study is crucial for improving our understanding of the interactions between the seismic cycle and fluid circulation. It deserves publication in a scientific journal. However, a comprehensive and in-depth analysis of some aspects related to the "normal" hydrogeological cycle is necessary to eliminate doubts and uncertainties in the interpretation of the data.

The pre-earthquake anomalies, if confirmed, are clear and well-evident.

Thank you for the encouraging comments.

So, I will now focus on the criticisms and issues to enhance the robustness of the study.

There is a complete absence of a description of the hydrogeological settings of the examined area and the geochemical characterization of the spring waters. It is necessary to place the analysed springs in a regional structural-geological and hydrogeological context.

Unfortunately, hydrogeological context of the spring water areas is not available.

The study's main weakness is the lack of data from March to September 2022 and historical data predating March 2022. Having data for this period could significantly enhance the comprehensiveness and depth of the analysis. The analysis of the historical time series of the spring could suggest a seasonal pattern. The data for March 2023 is very similar to that of March 2022. What happens in September and October 2023? The historical data from August 29, 2012, is insufficient to exclude cyclical and seasonal behaviour like that of 2022. It would be useful to obtain historical data within the same time interval where anomalies were recognized or extend the time series throughout 2023.

We mentioned (**on Page 14. Lines 347-348**) information for major ion contents for the water sample analyzed 10 years ago in 2012.

"We have also obtained a chemical analysis report on AYR water submitted with the business license application of the company dated 29 August 2012. The chemical analysis data of the samples collected more than 10 years ago include values only for $Na^+$, $Cl^–$, and $SO_4^{2–}$ as 3.86, 3.12, and 8.37 mg/l, respectively. These values are very close to the analysis result of the AYR water sample dated 8 March 2022 (AYR 1 which is the oldest sample in our data set) and the AYR water samples collected after 15 February (Table 2); confirming that these samples represent background values for the AYR spring water."

This was taken also as a proof that the March 2022 and March 2023 samples represent chemical background before manifestation of pre-earthquake crustal deformation and after the February 2023 Earthquakes earthquake, respectively. Seasonal change in the major ion content does not seem to exist because we have sample both from March (2022 and 2023) and August 2012 which have comparable content for at least Na+, Cl−, and SO4-2.  The copy of the business license issued on 29 August 2012 and chemical analysis report are given below.

Could the rainfall after earthquakes 1 and 2 have diluted the spring waters? What is the recharge/discharge cycle of this spring, and what are the factors that determine it?

Hydrogeological information is not available. In future, as we hope to be able initiate a detailed study in this area under a scope of a multi-disciplinary approach, we hope to be able to obtain some insights.

Suggestions for improvement:

1. Include a hydrogeological and hydrogeochemical framework of the area.

   Unfortunately, hydrogeological context of the study area is not available.

2. Enhance (if possible) Figure 2 with a circulation scheme and hydrogeological map.

   Because hydrogeological data are not available, this could not be possible at the moment. As we advance our knowledge through future studies, this will definitely be possible.

3. Include in the "Results and Discussion" section the geochemical characterization of the studied springs (e.g., Langelier Ludwig diagram - all samples).

   With EC and major ion analysis only, it is difficult to characterize water samples. But this will be possible in future studies.

4. The electrical conductivity values of Ayran appear to be shifting towards mixing with waters like those of Bahcepinar. Have you assessed this possibility?

   Bahçepınar water is confined in relatively surficial Alluvium deposit and we think that this water does not penetrate into lower lithologies as these are metamorphic and relatively impermeable.

5. The time axis length in Figure 3 is not consistent with time. Insert a break between March and September 2022 or adjust the proportions accordingly.

   We plotted water samples date on the x-axis in Figure 3 and the details are already given in Table 2.

6. Extend the time series to at least a year (using either new or historical data).

This is not possible because historical data (March-September 2022) are not available. New data can potentially be generated on new bottled samples but this requires further funding that is not available to us at present.

7. In Figure 3 Cont., clarify the meaning of "Average Rainfall."

   This is average daily rainfall. We have explained this in the revised manuscript.

8. Consider combining Figure 3 and Figure 3 Cont.

   We have tried this but the figure then becomes too long for a page. We leave this to Journal's production department to reformat as they wish.

It's challenging to hypothesize interpretative models with data from just one spring. It is necessary to exclude trends related to the hydrogeological cycle and then evaluate possible phenomena of mixing between different aquifers (e.g., shallow vs. deep) or variations in hydrodynamic properties related to the preparatory phases of the main seismic events. The analysis of stable isotopes of the water molecule could help understand potential mixing processes. Despite the samples not being acidified, an attempt to analyse trace elements should be made.

We agree. As more studies will be conducted in the future by us or others in the region, the referee's recommendations can be applied.

The physical mechanisms of the observed precursors are yet impossible to explain with certainty at this stage. In order to be able to suggest the mechanism(s) leading to the reported pre-earthquake geochemical anomalies, more work needs to be conducted; especially multi-disciplinary (seismological, geodetical, geochemical) and continuous earthquake monitoring networks must be established and run for a sufficiently long time

Although the referee's other comments have merits, as we have mentioned in the manuscript, we have been able to collect and analyze water samples from the earthquake region. In the region, no any other continuous or discrete data are available for pre-earthquake period. We hope that this paper ones published will pave the road for support of multi-disciplinary and multi-lateral project where all the issues raised by the referee can be studied.

We thank the anonymous referee for further comments and recommendations he/she has provided in a supplement file.

Below are **our responses in blue font** to the **referee's remarks in black fonts.**

The work tested the changes in electrical conductivity and the ion concentrations in two springs near the epicenters before and after two earthquakes (Mw 7.7 and Mw 7.6, in Kahramanmaraş), suggesting the anomalies with an increase electrical conductivity and the major ions before the earthquake. The analytical methods used and the results obtained are reliable, but the conclusions are limited due to the limited samples (the number of spring holes) and the significant results from only one point. The manuscript needs major revision. I believe that the authors have done their best to obtain data on all possible spring holes, and it would be difficult to recommend that they test more samples from other possible springs.

Thank you for the understanding. We concur with the referee that at the moment due to lack of any ongoing project and no funding, it is not possible to collect more samples from the area for analysis. On the other hand, we are almost certain that we cannot find bottled samples dated before the February earthquakes.

Therefore, from other perspectives, I think the following possible improvements still exist, depending on the authors.

1.      In terms of form, the contents of Table 2 and Figure 3 are somewhat repetitive, and it is suggested that the author could optimize or merge them.

Table 2 lists the data and Figure 3 shows the plot of the data for easy discussion and for the readers to follow.  This is conventional format to demonstrate.

2. Since the author only analyzed the relationship between the ion concentration changes and the earthquake from the perspective of time and location, this analysis is only correlation analysis and lacks causality analysis. Therefore, the author need try to find some data that can reflect the change of crust stress, such as deep drilling data, ground stress station data or satellite observation of surface displacement and deformation data (e.g., InSAR) to support the rationality of ion concentration changes from the perspective of time and space. I think this can greatly increase the reliability of the results of the work.

Referee's recommendations for utilizing additional data such as deep drilling, ground stress station data etc. are very logical but unfortunately unavailable in the study area for pre-earthquake time period. In the introduction Section of the manuscript, we mentioned a multidisiplinary continuous monitoring network (including borehole tiltmeter stations, GPS stations, soil radon stations, cold and warm spring water stations, microseismology stations) that was established in 2006 (Inan et al., 2007) and continued until 2012. Unfortunately, as we regretfully mentioned in the introduction

section, in 2012 the project was shut down and the montioring stations were removed. Therefore, we mentioned in the introduction that " the geosciences community was caught unprepared for the February earthquakes". We the geoscience community lost the opportunity of obtaining very valuable pre-earthquake data for these giant earthquakes in the region.

3. From the time series observation data of the spring water, the increase in ion concentration began as early as one year before the M7.7 earthquake. However, since the author only presented the results of one year, the process of increasing ion concentration was not fully displayed, and the audience could not clearly see when the increase in ion concentration began, and whether it was in a low value stable state more than one year before the M7.7 earthquake. Please add this part of data, if not, please explain the reason.

The oldest sample before the earthquake was dated March 2022 (about one year before the earthquake and for this sample no anomalies were present in EC and/or major ions content). Moeover, as we mentioned the major ions of this sample were very similar to chemical analysis results of a sample that was analyzed and submitted for Business licence apllication in 2012 ! So the sample for March 2022 represents background (no sign of crustal deformation). However, the samples from September 2022 until February 2023 (covering a duration of six months) show undisputed positive anomalies (increase) in major ions. Therefore, we mentioned that pre-earthquake anomaly lasted for at least six months (e.g., starting September 2022 and continuing until the earthquakes of 6 February after which the major ion contents started to diminish and in mid February background levels were reached.). There is a gap in samples between March 2022 and September 2022. So the anomaly could have started sometime after March 2022 and before September 2022; this would mean precursory anomaly more than six months. We have no way to speculate on this.

4. In addition, I would like to know whether the author has obtained synchronized spring temperature data, which I think is also crucial to reflect the process and results of underground fluid migration. If so, it is suggested that the author add relevant content and make analysis. If not, I suggest that the author refer to the schematic diagram in Figure 10 of this paper (He A, Singh R P. Groundwater level response to the Wenchuan earthquake of May 2008[J]. Geomatics, Natural Hazards and Risk, 2018.), and combine the location and lithology of the spring in your work to analyze and discuss the reasons for the abnormal spring.

The water samples we analyzed were commercially bottled samples sold at markets/shops. Therefore, any variations in spring water temperatures are not possible to comment on. If the multidisciplinary monitoring network has not been removed, then continuous water monitoring stations would have collected Ec, and temperature on

hourly basis. For example for Please refer to results of continuous monitoring of water stations by the authors (Inan et al., 2010) cited in the manuscript. Unfortunately, before February earthquakes there was no and yet there is no continuous water monitoring in the region.

Thank you for recommending work of He and Singh (2018), we have referred to this in the introduction of the revised manuscript. He and Singh (2018) paper discusses co-seismic response of the groundwater levels which is a well-known phenomenon reported to have taken place before, during, and after several earthquakes worldwide.

5. Line 402, two immediate mechanisms was presented. Here the authors should elaborate them in detail as you can as possible, for example providing some schematic diagrams associating to the potential mechanisms. This is very important to understand the physical processes and increase the reliability of this work.

Two immediate possible mechanisms potentially causing chemical changes in the water samples discussed in the manuscript are quite different. One envisages mixing of different aquifers (pre-earthquake crustal dilatation theory) and the other mechanism suggests chemical corrosion of rocks by positive hole currents causing weathering of the rock surface yielding more ions into the circulating water before earthquakes. The references given for each mechanism contain wealth of drawings; therefore, we did not want to repeat what others have already demonstrated.

**Referee 4: Dr. Vivek Walia**

Below are **our responses in blue font** to the **referee's remarks in black fonts.**

**CC2**: 'Comment on nhess-2023-133', Vivek Walia, 03 Oct 2023
The authors have presented an interesting study on fluctuations in ion concentrations in natural spring water as a precursor for the devastating $M_w$ 7.7 and $M_w$ 7.6 Kahramanmaraş (Türkiye) earthquakes of 6 February, 2023. The novelty of the study rests in the fact that despite the absence of continuous geochemical monitoring in the region, the authors have found a way to look for precursory anomalies indirectly, studying commercially available bottled water collected from natural springs close to the epicenters of the earthquakes. The manuscript is structured well and the main idea is presented clearly. I would recommend acceptance of the article after some revision.

Thank you for kind consideration and scientific support you have provided for this manuscript.

General comments:

- The manuscript should be checked thoroughly and inconsistencies in grammar, tenses of verbs, prepositions, etc. should be corrected.

- There are some prominent repetitions of entire sentences or parts of sentences in Abstract, Conclusions, and the last paragraph Result and Discussion. This should be edited.

Specific comments are mentioned within the manuscript.

We thank Dr. Vivek Walia for his encouragement and constructive view and review on our manuscript. We have revised the manuscript and made all the style and linguistic corrections and eliminated repetitions as the referee has suggested.

**Referee #5: Anonymous**

Below are **our responses in blue font** to the **referee's remarks in black fonts.**

**RC5**: 'Comment on nhess-2023-133', Anonymous Referee #4, 12 Oct 2023

The paper "Spring water anomalies before two consecutive earthquakes (Mw 7.7 and Mw 7.6) in Kahramanmaraş (Türkiye) on 6 February 2023" is of scientific relevance as it addresses the issue of hydrogeochemical precursors of major earthquakes. Although, unfortunately, two major earthquakes occurred this year in Turkey, within a short time of each other, causing massive destruction, the scientific community mobilized in an exemplary way in analyzing the situation and one of its efforts focused on precursors. A positive fact that could be deduced from these major occurrences is that it could calibrate the certain ionic anomalies that appear in natural spring waters about 6 months pre-earthquakes, as the authors point out in this paper.

As the article have very carefully considered by my fore-referees, I would like to draw attention to one aspect. In Subchapter 3.4, there is a discussion about the deduction of maximum distance from the epicenter where the anomalies could be detected versus magnitude. Actually, as it stated, an earthquake of Mw 4.5 could cause an ions anomaly at a distance of 100 km from the epicenter (Ayran Spring), but an earthquake of Mw 7,7, too.

I think there has to be a discussion in this paper or underlined that it would be addressed in the future, how to discriminate between a medium to small earthquake and a major one, if both could cause hydrogeochemical anomalies in the spring waters, because in the first case there really is no need to trigger an alarm.

We thank the referee for kind and open-minded approach and rightful comments that provided us an opportunity to discuss the issue a bit further. Parts of the reply below has been incorporated into the revised manuscript.

We agree that the precursory anomaly and (duration of this anomaly) that can be detected at a monitoring station depends on magnitude of earthquake and also the epicentral distance of the earthquake to the monitoring site. The closer the epicenter of an earthquake of a given magnitude will cause longer duration of anomaly at monitoring site. For example, taking Sultankhodhaev's (1984) empirical relation (log (DT) = 0.63 * M – b) which considers 1) earthquake magnitude (M), 2) distance of the earthquake epicenter to monitoring site (D), and 3) duration of anomaly (T), we can estimate duration of anomaly that can be expected in a monitoring site.

Taking the example given by the referee, it can be said that an earthquake of magnitude 4.5 occurring at a distance about 100 km to the monitoring station can lead to duration of anomaly (T) as about 5 days before the earthquake (using log(100*T) = 0,63* 4.5 - 0.15) . On the other hand, the duration of anomaly (T) for an earthquake Magnitude 7.7 whose epicenter is 100 km away from the monitoring site can be estimated as 295 days (using

log(100*T) = 0,63* 7.7 - 0.15). So, the greater the earthquake magnitude, the longer the duration of the precursory anomaly.

Obviously, we should keep in mind that the relations proposed by Dobrovolsky et al. (1979), Sultankhodhaev (1984), and Rikitake (1987) assume homogenous and isotropic crust where pre-earthquake stress and resultant strain propagates in all directions. In fact, we know that this assumption is not totally correct as microplates and/or block boundaries hinder stress transfer (e.g., Inan et al. 2012b). This issue should be seriously considered and care should be exercised.

The referee's last comment is also very worth to comment on. "How to discriminate between a medium to small earthquake and a major one" based on precursory anomalies would definitely require an ideal network of distributed monitoring stations (preferably of multi-disciplinary nature) established at sufficient distance to each other. So that each monitoring station would detect precursory anomalies for different durations and that would ideally help the interpreter to use simple algorithms to detect possible location and magnitude of an approaching earthquake. Here we should emphasize again that crustal heterogeneity and structural complexity is a must to consider.